# Nano-scale solution of the Poisson-Nernst-Planck (PNP) equations in a fraction of two neighboring cells reveals the magnitude of intercellular electrochemical waves

Karoline Horgmo Jæger[1]*, Ena Ivanovic[2], Jan P. Kucera[2], Aslak Tveito[1]

**1** Simula Research Laboratory, Oslo, Norway, **2** Department of Physiology, University of Bern, Bern, Switzerland

* karolihj@simula.no

**Data Availability Statement:** All code written in support of this publication is publicly available on

## Abstract

The basic building blocks of the electrophysiology of cardiomyocytes are ion channels integrated in the cell membranes. Close to the ion channels there are very strong electrical and chemical gradients. However, these gradients extend for only a few nano-meters and are therefore commonly ignored in mathematical models. The full complexity of the dynamics is modelled by the Poisson-Nernst-Planck (PNP) equations but these equations must be solved using temporal and spatial scales of nano-seconds and nano-meters. Here we report solutions of the PNP equations in a fraction of two abuttal cells separated by a tiny extracellular space. We show that when only the potassium channels of the two cells are open, a stationary solution is reached with the well-known Debye layer close to the membranes. When the sodium channels of one of the cells are opened, a very strong and brief electrochemical wave emanates from the channels. If the extracellular space is sufficiently small and the number of sodium channels is sufficiently high, the wave extends all the way over to the neighboring cell and may therefore explain cardiac conduction even at very low levels of gap junctional coupling.

## Author summary

Mathematical models are extensively used to understand how the electrochemical wave traversing the myocardium in every heartbeat work and fail, and how failures can be repaired. The models are based on averaging over small length-scales and traditionally represent cardiac tissue on the millimeter level. More recently, cell-based models have been developed representing the individual cardiomyocytes on the micro-meter scale. The cell-based models can accurately simulate electrochemical processes inside and around individual cells, but fail to accurately represent the strong gradients in the electrical potential and the ion concentrations very close to the cell membranes, referred to as the Debye layer. In order to resolve these gradients in a simulation, it is necessary to solve the Poisson-Nernst-Planck (PNP) equations and the solution of these equations requires a

Zenodo at https://doi.org/10.5281/zenodo.7572188.

**Funding:** KHJ and AT were supported by the Research Council of Norway via FRIPRO grant agreement #324239 (EMIx) and INTPART grant agreement #322312 (SIMBER), in addition to the SUURPh program funded by the Norwegian Ministry of Education and Research. JPK and EI were supported by Swiss National Science Foundation, grant number 310030_184707. The funders had no role in study design, data collection and analysis, decision to publish, or preparation of the manuscript.

**Competing interests:** The authors have declared that no competing interests exist.

mesh resolution on the order of nano-meters, thus strongly limiting the spatial domain that can be simulated. Here, we solve the PNP equations in a small domain including parts of two neighboring cells and the extracellular space between the cells. Our main observation is that the action potential of one cell can induce electrochemical waves that are sufficiently strong to affect a sufficiently close neighboring cell. This confirms that ephaptic coupling between cardiomyocytes is theoretically possible.

## 1 Introduction

At the tissue level ($\sim$mm), cardiac electrophysiology is properly represented using the well-established bidomain or monodomain models; see, e.g., [1]. However, a major limitation of the bidomain and monodomain models is that the cardiomyocytes are not explicitly present in the models. This limits the modeling capabilities of these models. At the level of individual cardiomyocytes ($\sim \mu$m), cell-based (EMI) models representing both the extracellular (E) space, the cell membrane (M) and the intracellular (I) space can be applied; see, e.g., [2–5]. The EMI models increase the modeling capabilities, allowing parameters to vary between and within individual cardiomyocytes, at the cost of significantly increased computing efforts needed to solve the equations; see, e.g., [6]. Yet, even though individual ion channels can be represented in the EMI model framework, the model does not represent the strong gradients in ion concentrations close to these channels. In order to study the electrodiffusion close to active ion channels placed in the cell membrane, it is necessary to solve the Poisson-Nernst-Planck equations (PNP, see, e.g., [7, 8]) which models electrodiffusion at $\sim$nm level. These equations are challenging to solve numerically because very strong gradients necessitate extremely fine spatial ($\sim$0.5 nm) and temporal ($\sim$0.01 ns) resolutions. The PNP equations can be simplified by assuming electroneutrality (charges sum up to zero everywhere in the computational domain), see, e.g., [9–13]. However, electroneutrality does not necessarily hold close the outlet (or inlet) of active ion channels and it remains unclear what consequences follow from assuming electroneutrality. A nice summary of the validity of different models close to the cell membranes is provided in [14] and limitations of electroneutrality are analyzed in [15].

Conventionally, electrochemical coupling between neighboring ventricular cardiomyocytes is believed to occur via gap junctions (GJ) that provide low-resistance pathways from cell to cell using the Connexin-43 (Cx43) protein, see, e.g., [16, 17]. An alternative route of conduction from cell to cell is via the extracellular space, referred to as ephaptic coupling. This alternative has been discussed for a very long time (see, e.g., [18, 19]) and recent modeling results, emphasizing the importance of the localization of the ion channels, indicate that ephaptic coupling is a viable alternative way of conduction, see, e.g., [3, 20–23]. A related discussion goes on in computational neurophysiology where the question is whether or not a neuron can set off an excitation wave in a neighboring cell; see [24–28]. One inherent difficulty in elucidating these questions is that the space between cells can be very small ($\sim$nm), making it very difficult to measure the electrical potential without perturbing its dynamics by the measuring device; see, e.g., [29].

The purpose of our report is to present solutions of the PNP equations between and in a fraction of two neighboring cells. In the simulations, we observe very strong spatial ($\sim$ 50 mV/nm) and temporal ($\sim$ 25 mV/ns) gradients in the electrical potential and this implies that we need to use an extremely fine mesh and very small time steps. We are thus only able to simulate a very small fraction of the cells and the extracellular space between them ($\sim$ 0.2 $\mu$m$^3$) for a very brief time interval ($\sim$ 1 $\mu$s). For comparison, the volume of a ventricular myocyte is about

30,000 $\mu$m$^3$ and the upstroke of the action potential takes about 1 ms. On the other hand, the cross-sectional area of an ion channel is about 4 nm$^2$ (see [30]) and we are able to represent that area scale in our simulations.

When only the potassium channel cluster is open in both cells, a stationary solution is reached with the well-known Debye layer close to both cell membranes. From this stationary solution, we open the sodium channels in the pre-junctional cell. This leads to a very strong electrochemical wave emanating from the sodium channels of the pre-junctional cell. The strength of this wave depends on the number of sodium channels in the cluster of channels under consideration and also on the width of the extracellular space. When the extracellular space is sufficiently small and the number of sodium channels is sufficiently high, the wave emanating from the sodium channels of the pre-junctional cell, reaches the post-junctional cell and can be strong enough to set off an excitation in the post-junctional cell.

## 2 Methods

### 2.1 The Poisson-Nernst-Planck (PNP) equations

We perform simulations of a small part of an intercalated disc. We consider a computational domain consisting of a fraction of two intracellular domains, with two associated membrane domains and an extracellular domain located between the membrane domains, as illustrated in Fig 1A. In the membrane domains, clusters of K$^+$ and Na$^+$ channels are embedded, as illustrated for a single K$^+$ and Na$^+$ channel in Fig 1B.

**2.1.1 Parameter values.**   In this domain, the electric potential and the concentration of the four ionic species Na$^+$, K$^+$, Ca$^{2+}$, and Cl$^-$ are modeled using the Poisson-Nernst-Planck (PNP) model:

$$\nabla \cdot (\varepsilon_r \varepsilon_0 \nabla \phi) = -(\rho_0 + F\sum_k z_k c_k), \tag{1}$$

$$\frac{\partial c_k}{\partial t} = \nabla \cdot D_k \nabla c_k + \nabla \cdot \left( \frac{D_k z_k e}{k_B T} c_k \nabla \phi \right), \tag{2}$$
$$\text{for } k = \{\text{Na}^+, \text{K}^+, \text{Ca}^{2+} \text{ and } \text{Cl}^-\}.$$

Here, $\phi$ is the electric potential (in mV) and $c_k$ are the ion concentrations (in mM). Furthermore, $F$ is Faraday's constant (in C/mol), $\varepsilon_0$ is the vacuum permittivity (in fF/m), $\varepsilon_r$ is the (unitless) relative permittivity of the medium, $\rho_0$ is the background charge density (in C/m$^3$), $D_k$ are the diffusion coefficients for each ion species (in nm$^2$/ms), $e$ is the elementary charge (in C), $k_B$ is the Boltzmann constant (in mJ/K), $T$ is the temperature (in K), and $z_k$ are the (unitless) valences of the ion species. Furthermore, time is given in ms and length is given in nm. Note that in the vizualizations below, we use ns as the temporal unit because of the extreme swiftness of the dynamics involved.

Defining

$$\varepsilon = \varepsilon_r \varepsilon_0, \tag{3}$$

$$\rho = \rho_0 + F\sum_k z_k c_k, \tag{4}$$

$$\beta_k = \frac{z_k e}{k_B T}, \tag{5}$$

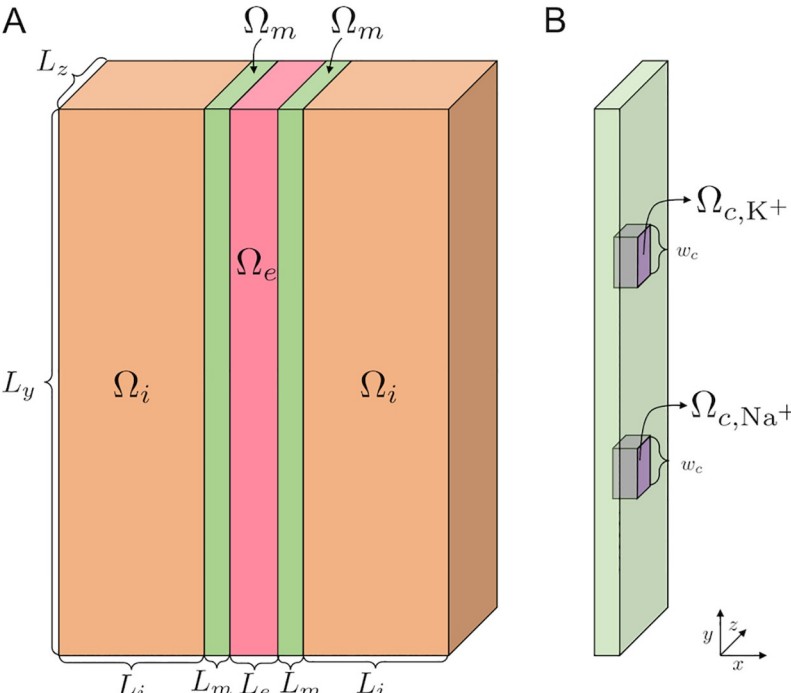

**Fig 1. Illustration of the computational domain. A**. Illustration of the domain. We consider two intracellular domains $\Omega_i$ (orange), two membrane domains $\Omega_m$ (green) and an extracellular domain $\Omega_e$ (pink). The left cell is referred to as the pre-junctional cell and the right cell is referred to as the post-junctional cell. **B**. Illustration of the channels embedded in the membrane, $\Omega_{c,K^+}$ and $\Omega_{c,Na^+}$ (purple). The width of the channels is $w_c$ in the $y$- and $z$-directions. Note that in our simulations, we typically consider clusters of channels, as illustrated in Fig 2. Note also that the illustrations in this figure are not drawn in scale, but the values of the lengths defining the geometry used in the simulations are specified in Table 1.

the system can be rewritten more compactly as

$$\nabla \cdot (\varepsilon \nabla \phi) = -\rho, \tag{6}$$

$$\frac{\partial c_k}{\partial t} = \nabla \cdot D_k \nabla c_k + \nabla \cdot (D_k \beta_k c_k \nabla \phi), \tag{7}$$

$$\text{for } k = \{Na^+, K^+, Ca^{2+} \text{ and } Cl^-\}.$$

**2.1.2 Initial conditions and background charge density, $\rho_0$.** The initial conditions for the ion concentrations in the intracellular and extracellular domains are specified in Table 2.

**Table 1. Default geometry parameter values used in the simulations (see Fig 1).**

| Parameter | Value |
| --- | --- |
| $L_i$ | 1000 nm |
| $L_m$ | 5 nm |
| $L_e$ | 5–30 nm |
| $L_y$ | 300 nm |
| $L_z$ | 300 nm |
| $w_c$ | 1 nm |

**Table 2. Initial conditions for the ion concentrations.** The values are based on [11, 12].

| Ion | Intracellular | Extracellular |
| --- | --- | --- |
| $Na^+$ | 12 mM | 100 mM |
| $K^+$ | 125 mM | 5 mM |
| $Ca^{2+}$ | 0.0001 mM | 1.4 mM |
| $Cl^-$ | 137.0002 mM | 107.8 mM |

In the membrane (outside ion channels), all ion concentrations are set to zero. Furthermore, in the ion channels, the concentration of the ion species that is able to move through the channel is initially set up to vary linearly from the intracellular to the extracellular part of the channel, and the remaining ion concentrations are set to zero.

The background charge density, $\rho_0$, is set up such that the entire domain is electroneutral at $t = 0$. This means that $\rho_0 = 0$ everywhere except in the ion channels, where it is determined by the initial conditions for the ion species able to move through the channel.

In S1 Appendix, we show the simulation results for a different choice of $\rho_0$ than the default linear profile. More specifically, we consider initial conditions in the channels specified by a jump from the intracellular to the extracellular concentration in the center of the channel. The main results concerning the magnitude of the changes of the extracellular potential between the cells seem to be quite similar for this other choice of $\rho_0$ in the channels. However, some differences between the solutions are observed (see S1 Appendix).

In S1 Appendix, we also show the results of simulations with a different choice of initial conditions for the intracellular concentrations. More specifically, the intracellular $Na^+$ concentration in cardiomyocytes is known to vary between species, with a concentration of about 4–8 mM for most mammalian species, but 10–15 mM in rat and mouse [31–34]. Thus, the default value of the intracellular $Na^+$ concentration reported in Table 2 (12 mM) is mostly representative of rat or mouse cardiomyocytes, and a value of about 8 mM would be more realistic for healthy human cardiomyocytes [32]. In S1 Appendix, we compare the results of simulations using either 12 mM or 8 mM as initial conditions for the intracellular $Na^+$ concentration. In addition, the intracellular $Cl^-$ concentration is set to either 137.0002 mM or 133.0002 mM, respectively, in order to maintain electroneutrality at $t = 0$. In S1 Appendix, we observe that the two choices of initial conditions give very similar results.

The default geometry used in the simulations is specified in Fig 1 and Table 1. In the purely intracellular and extracellular spaces, the relative permittivity, $\varepsilon_r$, is set to $\varepsilon_1 = 80$. In the membrane and the ion channels, $\varepsilon_r$ is set to $\varepsilon_m = 2$. Furthermore, in the purely intracellular and extracellular domains, $\Omega_i$ and $\Omega_e$, the diffusion coefficients for the ions are as specified in Table 3. In the membrane domains, $\Omega_m$, all diffusion coefficients are set to zero. In the $K^+$ channels, $\Omega_{c,K^+}$, the diffusion coefficient for $K^+$ is set to $d_{K^+} \times D_{K^+}$, where $d_{K^+}$ is a channel scaling factor. The diffusion coefficient for the remaining ions are set to zero in the $K^+$ channels. Similarly, in the $Na^+$ channels, $\Omega_{c,Na^+}$, all diffusion coefficients are set to zero when the channel is closed and the diffusion coefficient for $Na^+$ is set to the value $d_{Na^+} \times D_{Na^+}$ when the channel is open. The justification for the choice of scaling factors $d_{K^+}$ and $d_{Na^+}$ is specified below.

**2.1.3 Parameterization of the channels.** **Channel area and diffusion**. We let the cross sectional area of each ion channel be 1 nm × 1 nm. In addition, we adjust the value of the scaling factor $d_{Na^+}$ for the diffusion through an open $Na^+$ channel so that the peak current through an open channel is approximately 2 pA, based on single channel measurements from [37]. Furthermore, based on estimation from [38] that the conductance of $I_{K1}$ channels is about 4 times smaller than the conductance of $I_{Na}$ channels and that the number of $I_{Na}$ channels is about 5

**Table 3. Parameter values used in the simulations.**

| Parameter | Description | Value | Ref. |
|---|---|---|---|
| $F$ | Faraday's constant | 96485.3365 C/mol | [35] |
| $\varepsilon_0$ | Vacuum permittivity | 8854 fF/m | [35] |
| $\varepsilon_1$ | Relative permittivity, $\varepsilon_r$, in $\Omega_i$ and $\Omega_e$ | 80 | [36] |
| $\varepsilon_m$ | Relative permittivity, $\varepsilon_r$, in $\Omega_m$, $\Omega_{c,Na^+}$ and $\Omega_{c,K^+}$ | 2 | [36] |
| $D_{Na^+}$ | Diffusion coefficient for $Na^+$ in $\Omega_i$ and $\Omega_e$ | $1.33 \cdot 10^6$ nm²/ms | [11] |
| $D_{K^+}$ | Diffusion coefficient for $K^+$ in $\Omega_i$ and $\Omega_e$ | $1.96 \cdot 10^6$ nm²/ms | [11] |
| $D_{Ca^{2+}}$ | Diffusion coefficient for $Ca^{2+}$ in $\Omega_i$ and $\Omega_e$ | $0.71 \cdot 10^6$ nm²/ms | [11] |
| $D_{Cl^-}$ | Diffusion coefficient for $Cl^-$ in $\Omega_i$ and $\Omega_e$ | $2.03 \cdot 10^6$ nm²/ms | [11] |
| $d_{K^+}$ | Scaling factor for $D_{K^+}$ in $\Omega_{c,K^+}$ | 0.025 | |
| $d_{Na^+}$ | Scaling factor for $D_{Na^+}$ in $\Omega_{c,Na^+}$ | 0.5 | |
| $z_{Na^+}$ | Valence of $Na^+$ | 1 | |
| $z_{K^+}$ | Valence of $K^+$ | 1 | |
| $z_{Ca^{2+}}$ | Valence of $Ca^{2+}$ | 2 | |
| $z_{Cl^-}$ | Valence of $Cl^-$ | −1 | |
| $e$ | Elementary charge | $1.60217662 \cdot 10^{-19}$ C | [35] |
| $k_B$ | Boltzmann constant | $1.380649 \cdot 10^{-20}$ mJ/K | [35] |
| $T$ | Temperature | 310 K | |

times larger than the number of $I_{K1}$ channels, the scaling factor for the $K^+$ channel diffusion, $d_{K^+}$, is set to be 20 times smaller than $d_{Na^+}$.

**Representation of channel clusters.** $Na^+$ channels have been found to form clusters on the membrane of cardiomyocytes (see, e.g., [39–41]). In order to investigate how such channel clustering affects the simulation results, we consider some cases of different numbers of open $Na^+$ channels in a cluster. The clusters are represented in the model by increasing the width, $w_c$ (and consequently the area) of the $Na^+$ channels that are embedded in the membrane (see Fig 1B). The scaling factor for the $Na^+$ channel diffusion, $d_{Na^+}$ is kept constant when the channel cluster size is increased. In our simulations, we consider $Na^+$ channel clusters consisting of 4, 36, 196 or 900 $Na^+$ channels (see Fig 2).

For the $K^+$ channel clusters, we consider a cluster of 36 channels for each of the two cells. When the size of the $Na^+$ channel clusters are adjusted, we adjust the scaling factor for the $K^+$ channel diffusion, $d_{K^+}$, by a factor $\frac{N_{Na}}{N_K}$, where $N_{Na}$ is the number of $Na^+$ channels and $N_K = 36$ is the number of $K^+$ channels. This is done to maintain the ratio of the magnitude of the $I_{Na}$ and $I_{K1}$ currents.

## 2.2 Simulation setup

Using the model setup described in Section 2.1, we perform simulations to investigate possible ephaptic coupling between the two cells. The simulations are separated into two parts:

**Part 1: Closed $Na^+$ channels.** We start the simulation from electroneutral conditions (see Section 2.1.2) with physiological values for the extracellular and intracellular ion concentrations (see Table 2). We then let the model approach a resting state by letting only the $K^+$ channels be open between the intracellular and extracellular spaces on the membrane of both cells. For the initial conditions (electroneutral), $\rho$ is zero, and $\phi$ is constant (see (6) and (10)). We run this simulation until we seem to have reached a steady-state solution for open $K^+$ channels. More specifically, we run the simulation until the change in the computed transmembrane

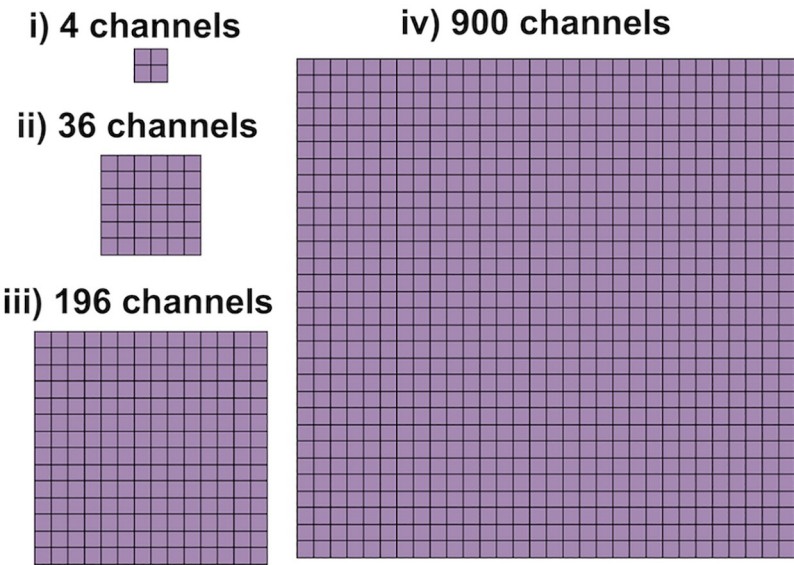

**Fig 2. Illustration of the four different Na$^+$ channel cluster sizes considered in our simulations.** In the case of 4 channels, the channel cluster width is 2 nm, whereas in the case of 36, 196 and 900 channels, the width of the channel clusters are 6 nm, 14 nm and 30 nm, respectively.

potential (see Section 2.4.2) is reduced by a factor of at least 500 compared to the maximal change in the transmembrane potential occurring in the beginning of the simulation.

**Part 2: Open Na$^+$ channel cluster on the membrane of the pre-junctional cell**. After the simulation in Part 1, we simulate the opening of the Na$^+$ channel cluster on the membrane of the pre-junctional cell and observe how this affects the ion concentrations and electric potential in the extracellular space between the cells. In particular, we wish to investigate whether the Na$^+$ channel opening on the pre-junctional cell seems to affect the electric potential outside the post-junctional cell enough to potentially initiate Na$^+$ channel opening on the membrane of the post-junctional cell. The K$^+$ channels are kept open in this part of the simulation. The Na$^+$ channels on the pre-junctional cell are kept open until the transmembrane potential, $v$, across the Na$^+$ channel cluster of the pre-junctional cell reaches a value of 30 mV. See Section 2.4.2 for the definition of $v$.

## 2.3 Boundary conditions

In order to simulate possible ephaptic coupling between two neighboring cells, we would like to solve the PNP system in a volume consisting of two cells and the surrounding extracellular space (i.e., about 60,000 $\mu$m$^3$). If we, for simplicity, assume that every computational mesh block is 1 nm$^3$, this would lead to a computational problem with $6 \times 10^{13}$ blocks, which is currently out of reach. We therefore need to restrict the problem spatially and that means that we need to define boundary conditions at locations where there are no actual physical boundaries. Of course, we want to define such boundary conditions in a way that provides the best possible model of the physiology under consideration. We will deal with combinations of Neumann and Dirichlet conditions and present results for two alternative sets of boundary conditions (one alternative will be shown only in S1 Appendix).

**2.3.1 Natural boundary conditions for the PNP system.** For the Poisson Eq (6), a compatibility condition needs to be satisfied in order to obtain proper solutions of the equation,

see, e.g., [42–44]. This condition follows from Gauss' theorem after integration over the entire computational domain,

$$-\int_\Omega \rho \; dV = \int_{\partial\Omega} \varepsilon \frac{\partial \phi}{\partial n} \; dS. \tag{8}$$

In the case of *natural* boundary conditions, i.e., when the normal derivatives of the both the electrical potential and the all the concentrations vanishes at the boundary, this condition holds. First, the right-hand side is clearly zero (by the boundary condition on the electrical potential). Also, if we assume that the ions initially are in an electroneutral state in the whole domain, then the integral of $\rho(\cdot, t = 0)$ is zero. Furthermore, this integral does not change in time because the integral of each specie is constant,

$$\begin{aligned}
\frac{d}{dt}\int_\Omega c_k(\cdot, t) \; dV &= \int_\Omega \frac{\partial c_k(\cdot, t)}{\partial t} \; dV \\
&= \int_\Omega \nabla \cdot (D_k \nabla c_k + (D_k \beta_k c_k \nabla \phi)) \; dV \\
&= \int_{\partial\Omega} \left( D_k \frac{\partial c_k}{\partial n} + D_k \beta_k c_k \frac{\partial \phi}{\partial n} \right) dS \\
&= 0,
\end{aligned} \tag{9}$$

for any $k = \{Na^+, K^+, Ca^{2+}, Cl^-\}$. Since both the left-hand side and the right-hand side of (8) are zero, the compatibility condition is satisfied for natural boundary conditions.

**2.3.2 Physiologically motivated boundary conditions.** The set of boundary conditions considered in most of our simulations is a physiologically motivated and relatively complex mix of Dirichlet and Neumann type boundary conditions. These boundary conditions are applied in all simulations reported in the Results section of the paper, and all simulations in S1 Appendix, unless otherwise stated.

In the physiologically motivated boundary conditions, we apply a Dirichlet boundary condition fixing the electric potential, $\phi$, at zero in an area of the boundary in the *y*- and *z*-directions in the extracellular cleft. More specifically, this area, $\partial\Omega_D^D$ is defined for the *x*-values corresponding to the middle third of the extracellular cleft, as illustrated in Fig 3. In $\partial\Omega_D^D$, we apply Dirichlet boundary conditions for the all the ionic concentrations as well. The concentrations in this area are fixed at the initial conditions for the extracellular space (see Table 2). At the leftmost boundary of the pre-junctional cell, marked as $\partial\Omega_{N,1}^D$ in Fig 3, we fix the concentrations at the initial conditions for the intracellular space see Table 2), and apply no-flux Neumann boundary conditions for $\phi$. At the rightmost boundary of the post-junctional cell, marked as $\partial\Omega_{N,2}^D$ in Fig 3, we use the same boundary conditions as for $\partial\Omega_{N,1}^D$ in the simulations used to find the resting state of the system (see Section 2.2). However, when the $Na^+$ channels of the pre-junctional cell are opened, we apply Dirichlet boundary conditions for $\phi$ on the right boundary of the post-junctional cell ($\partial\Omega_{N,2}^D$), fixing the potential at this boundary at the value of $\phi$ found at the end of the resting state simulations. On the remaining part of the domain boundary, $\partial\Omega_N^N$, we apply no-flux Neumann boundary conditions for both $\phi$ and the concentrations.

In summary, the boundary conditions are given by

$$\varepsilon\nabla\phi \cdot \mathbf{n} = 0 \qquad \text{at } \partial\Omega_N^N \cup \partial\Omega_{N,1}^D, \tag{10}$$

$$\varepsilon\nabla\phi \cdot \mathbf{n} = 0 \qquad \text{at } \partial\Omega_{N,2}^D \text{ for } t \leq t^*, \tag{11}$$

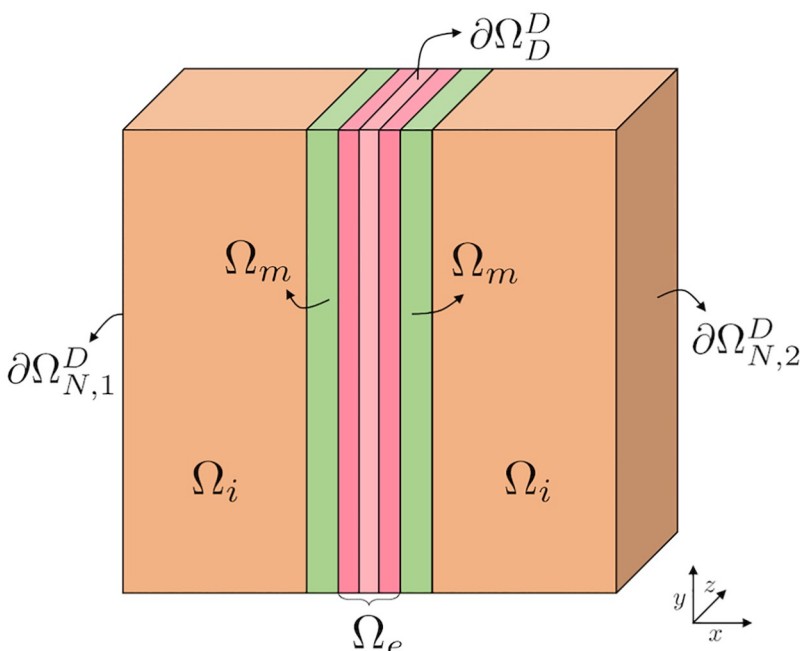

**Fig 3. Illustration of different parts of the domain boundary involved in the physiologically motivated boundary conditions (see Section 2.3.2) used in most of our simulations.** In $\partial\Omega_D^D$, i.e., in the boundary in the $y$- and $z$-directions for the $x$-values corresponding to the middle third of the extracellular cleft, we apply Dirichlet boundary conditions for both $\phi$ and all the ionic concentrations. In the simulations used to find the resting state of the system (see Section 2.2), we apply Neumann boundary conditions for $\phi$ and Dirichlet boundary conditions for the concentrations at $\partial\Omega_{N,1}^D$ and $\partial\Omega_{N,2}^D$. In the simulations with open Na$^+$ channels (see Section 2.2), we apply Neumann boundary conditions for $\phi$ and Dirichlet boundary conditions for the concentrations at $\partial\Omega_{N,1}^D$ and Dirichlet boundary conditions for both the concentrations and $\phi$ at $\partial\Omega_{N,2}^D$. In the remaining boundary $\partial\Omega_N^N = \partial\Omega \setminus (\partial\Omega_{N,1}^D \cup \partial\Omega_{N,2}^D \cup \partial\Omega_D^D)$, we apply Neumann boundary conditions for both $\phi$ and the ionic concentrations.

$$\phi = \phi(t^*) \qquad \text{at } \partial\Omega_{N,2}^D \text{ for } t > t^*, \tag{12}$$

$$\phi = 0 \qquad \text{at } \partial\Omega_D^D, \tag{13}$$

$$D_k \nabla c_k \cdot \mathbf{n} = 0 \qquad \text{at } \partial\Omega_N^N, \tag{14}$$

$$c_k = c_{k,i}^0 \qquad \text{at } \partial\Omega_{N,1}^D \cup \partial\Omega_{N,2}^D, \tag{15}$$

$$c_k = c_{k,e}^0 \qquad \text{at } \partial\Omega_D^D, \tag{16}$$

for $k = \{$Na$^+$, K$^+$, Ca$^{2+}$ and Cl$^-\}$, where $c_{k,e}^0$ and $c_{k,i}^0$ are the extracellular and the intracellular concentrations, respectively, of the ion species $k$ specified in Table 2 and $t^*$ is the point in time when the Na$^+$ channels of the pre-junctional cell are opened (see Section 2.2). Furthermore, $\partial\Omega_{N,1}^D$, $\partial\Omega_{N,2}^D$ and $\partial\Omega_D^D$ are illustrated in Fig 3, and $\partial\Omega_N^N$ is the remaining part of the domain boundary, defined by $\partial\Omega_N^N = \partial\Omega \setminus (\partial\Omega_{N,1}^D \cup \partial\Omega_{N,2}^D \cup \partial\Omega_D^D)$, where $\partial\Omega$ is the entire domain boundary.

Note that in S1 Appendix, we show the results of simulations where Neumann boundary conditions are applied at all parts of the boundary for all ionic concentrations and everywhere except at the right boundary for $\phi$. The results of the simulations with this different choice of boundary condition seem to be very close to the results obtained using the boundary conditions described above.

## 2.4 Numerical solution of the PNP equations

We solve the PNP model equations using an implicit finite difference discretization of a splitting scheme in which the two equations (6) and (7) are solved separately. That is, for each time step $n$, we assume that the concentrations $c_k^{n-1}$ (and consequently $\rho^{n-1}$) are known for $t_{n-1}$ and the PNP model equations are solved in two steps:

**Step 1**. Solve

$$\nabla_h \cdot (\varepsilon \nabla_h \phi^n) = -\rho^{n-1} \tag{17}$$

**Step 2**. Solve

$$\frac{c_k^n - c_k^{n-1}}{\Delta t} = \nabla_h \cdot D_k \nabla_h c_k^n + \nabla_h \cdot \left( D_k \beta_k c_k^n \nabla_h \phi^n \right),$$

$$\text{for } k = \{\text{Na}^+, \ \text{K}^+, \ \text{Ca}^{2+}, \ \text{Cl}^-\}, \tag{18}$$

where $\nabla_h$ refers to a finite difference discretization of $\nabla$. The finite difference discretizations used in Step 1 and Step 2 are described in detail in S1 Appendix.

**2.4.1 Mesh.** The spatial domain is discretized using a simple adaptive mesh in the $x$-, $y$- and $z$-directions. Close to the membrane and in the membrane, $\Delta x$ is set to 0.5 nm. Further away from the membrane, $\Delta x$ is doubled for each grid point. Similarly, $\Delta y$ and $\Delta z$ are set to 1 nm close to the ion channels and doubled for each grid point further away from the channels. An example grid is illustrated in the $(x, y)$- and the $(x, z)$-planes in Fig 4. This simple approach to create an adaptive mesh considerably reduces the number of mesh points and thus also the CPU efforts. For example, for the default domain with $L_y = L_z = 300$ nm, $L_i = 1000$ nm, and $L_e = 10$ nm, the number of grid points is reduced from about $364 \cdot 10^6$ for a uniform mesh with $\Delta x = 0.5$ nm and $\Delta y = \Delta z = 1$ nm to about 18,000 for the adaptive mesh. In other words, the number of unknowns to be computed for each time point is reduced by a factor of about 20,000.

In order to avoid unstable numerical solutions, we use the time step $\Delta t = 2 \cdot 10^{-8}$ ms = 0.02 ns.

**2.4.2 Definition of the membrane potential and Na$^+$ current.** In our simulations, the transmembrane potential is defined as

$$v = \phi(\mathbf{x}_i) - \phi(\mathbf{x}_e), \tag{19}$$

where $\mathbf{x}_i$ and $\mathbf{x}_e$ are the first grid points (in the $x$-direction) outside of the membrane in the intracellular and extracellular spaces, respectively, as illustrated in Fig 5A. Furthermore, the sodium current, $I_{\text{Na}}$, is defined by the associated flux,

$$I_{\text{Na}} = F A_{\text{Na}^+} D_{\text{Na}^+} \left( \frac{c_{\text{Na}^+}(\mathbf{x}_i) - c_{\text{Na}^+}(\mathbf{x}_e)}{\|\mathbf{x}_i - \mathbf{x}_e\|_2} + \beta_{\text{Na}^+} c_{\text{Na}^+}(\mathbf{x}_m) \frac{\phi(\mathbf{x}_i) - \phi(\mathbf{x}_e)}{\|\mathbf{x}_i - \mathbf{x}_e\|_2} \right), \tag{20}$$

where $\mathbf{x}_i$ and $\mathbf{x}_e$ are the first grid points (in the $x$-direction) inside of the Na$^+$ channel, close to the intracellular and extracellular spaces, respectively, and $\mathbf{x}_m$ is a point in the center of the channel as illustrated in Fig 5B. Moreover, $A_{\text{Na}^+}$ is the area of the membrane covered by the

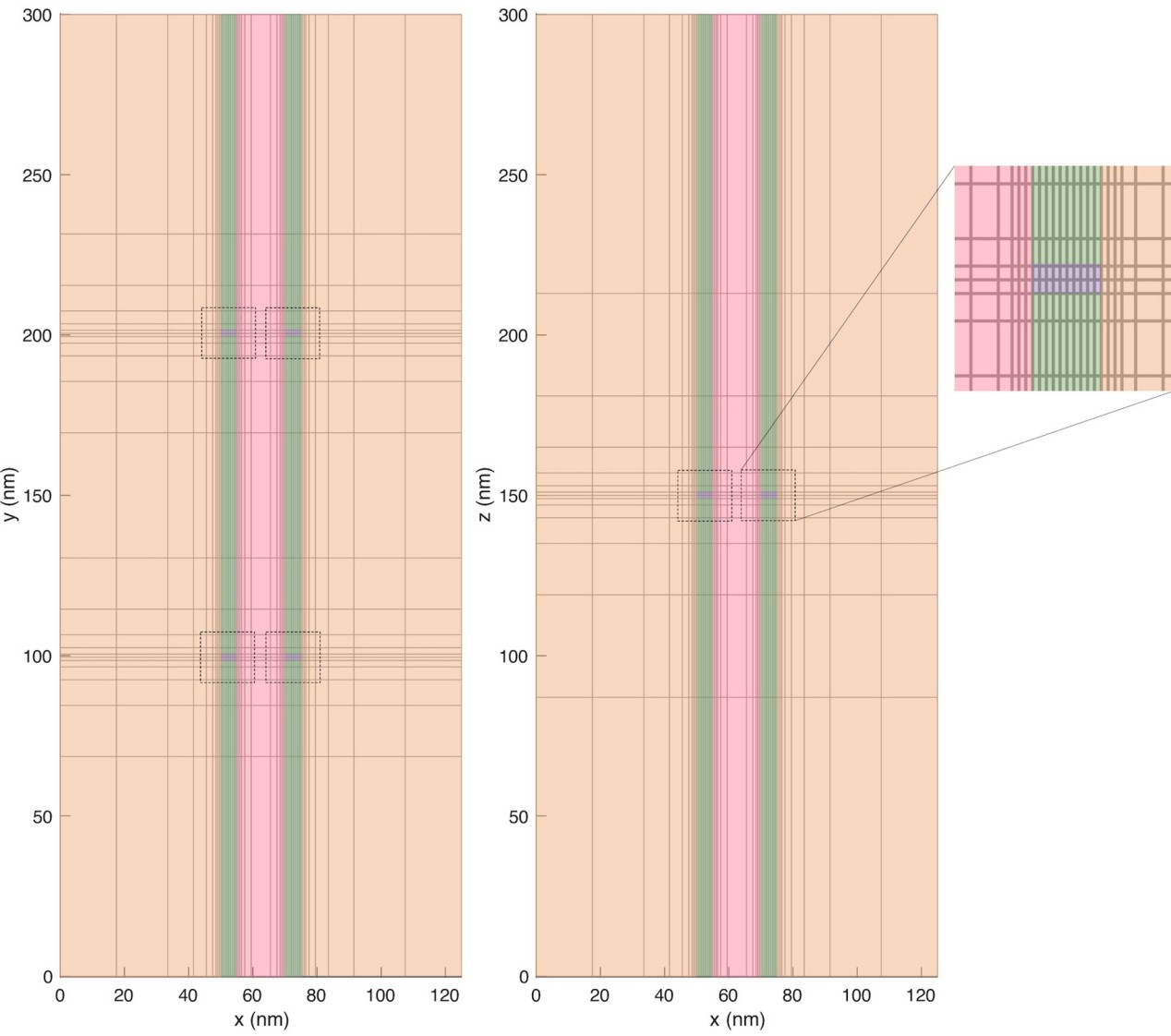

**Fig 4. Illustration of an example mesh for $L_e$ = 15 nm and $L_i$ = 50 nm in the $(x, y)$- and the $(x, z)$-planes.** The intracellular domain is colored orange, the extracellular domain is colored pink, the membrane is colored green and the ion channels are colored purple. In this figure, we consider K$^+$ and Na$^+$ channel clusters consisting of 2×2 channels of each type in the membrane of both cells. In the $(x, y)$-plane, the two types of ion channel are located as different locations (different $y$-values), whereas in the $(x, z)$-plane the two types of ion channel overlap because they are located at the same $z$-values. The mesh in the area close to the ion channels (indicated by dashed lines) is shown in more detail on the right side of the figure.

Na$^+$ channel cluster, and $\|\cdot\|_2$ is the Euclidean norm. The parameter $\beta_{\text{Na}^+}$ is defined in (5) and has a value of 0.037 mV$^{-1}$ for the parameter values specified in Table 3.

## 3 Results

### 3.1 The resting state of the PNP equations

Before investigating the dynamics involved with the opening of Na$^+$ channels on the membrane one of the two cells included in the simulations, we run simulations to obtain a resting state for the model, as described in Section 2.2. We run this type of simulation for three

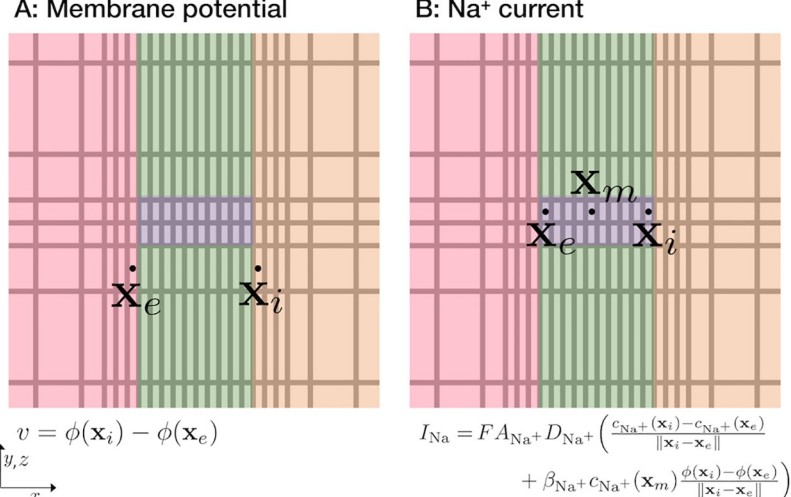

**Fig 5. Illustration of the definition of the transmembrane potential, $v$, and the Na$^+$ current, $I_{\mathrm{Na}}$, used in our computations.** The illustrations show a small part of the mesh in the $(x, y)$- or $(x, z)$-planes close to the Na$^+$ channel (as illustrated on the right side of Fig 4).

different lengths of the extracellular space between the cells ($L_e = 5$ nm, $L_e = 10$ nm, and $L_e = 30$ nm).

**3.1.1 The resting state depends on the size of the extracellular space.** Fig 6 shows the extracellular potential, $\phi$, the concentration of Na$^+$, K$^+$, Ca$^{2+}$, and Cl$^-$ ions and the charge density, $\rho$, in the extracellular space between the cells when the solutions seem to have reached steady state in the simulations performed to obtain the resting state of the system. In these simulations, only the K$^+$ channels are open on the membrane of the two cells. We show the solutions in a plane in the $x$- and $y$-directions in the center of the domain in the $z$-direction. We consider the solutions close to the K$^+$ channel, with the K$^+$ channels located in the center of the plots in the $y$-direction. Close to the membrane of both cells, we observe a boundary layer with slightly different values of the potential, $\rho$ and the concentrations than in the remaining extracellular space. This boundary layer extends further into the domain in the area close to the K$^+$ channels.

In Fig 7, we further illustrate the resting state solutions as lines in the $x$-direction through the K$^+$ channels and through a line across the main membrane. We observe that the electric potential $\phi$ is about $-80$ mV in the intracellular space and about $0$ mV in the extracellular space, resulting in a transmembrane potential of about $-80$ mV. We can also observe some small boundary layers in the ion concentrations close to the membrane. In the lower row of Fig 7, we show the charge density, $\rho$, computed from the ionic concentrations using (4). For the main membrane (dotted line), we clearly see that this value is close to zero everywhere, except in the boundary layer in both the intracellular and the extracellular spaces close to the membrane. In the line crossing the K$^+$ channel cluster, we also observe a non-zero charge density inside and close to the channel cluster. The profile of $\rho$ in the K$^+$ channel cluster is a consequence of small deviations at steady-state from the purely linear profile used for $\rho_0$ and initial condition for the K$^+$ in the cluster and is discussed further in S1 Appendix.

## 3.2 Opening the Na$^+$ channels on the pre-junctional cell

Starting from the solution at the end of the resting state simulations, we run simulations with open Na$^+$ channels in the membrane of the pre-junctional cell.

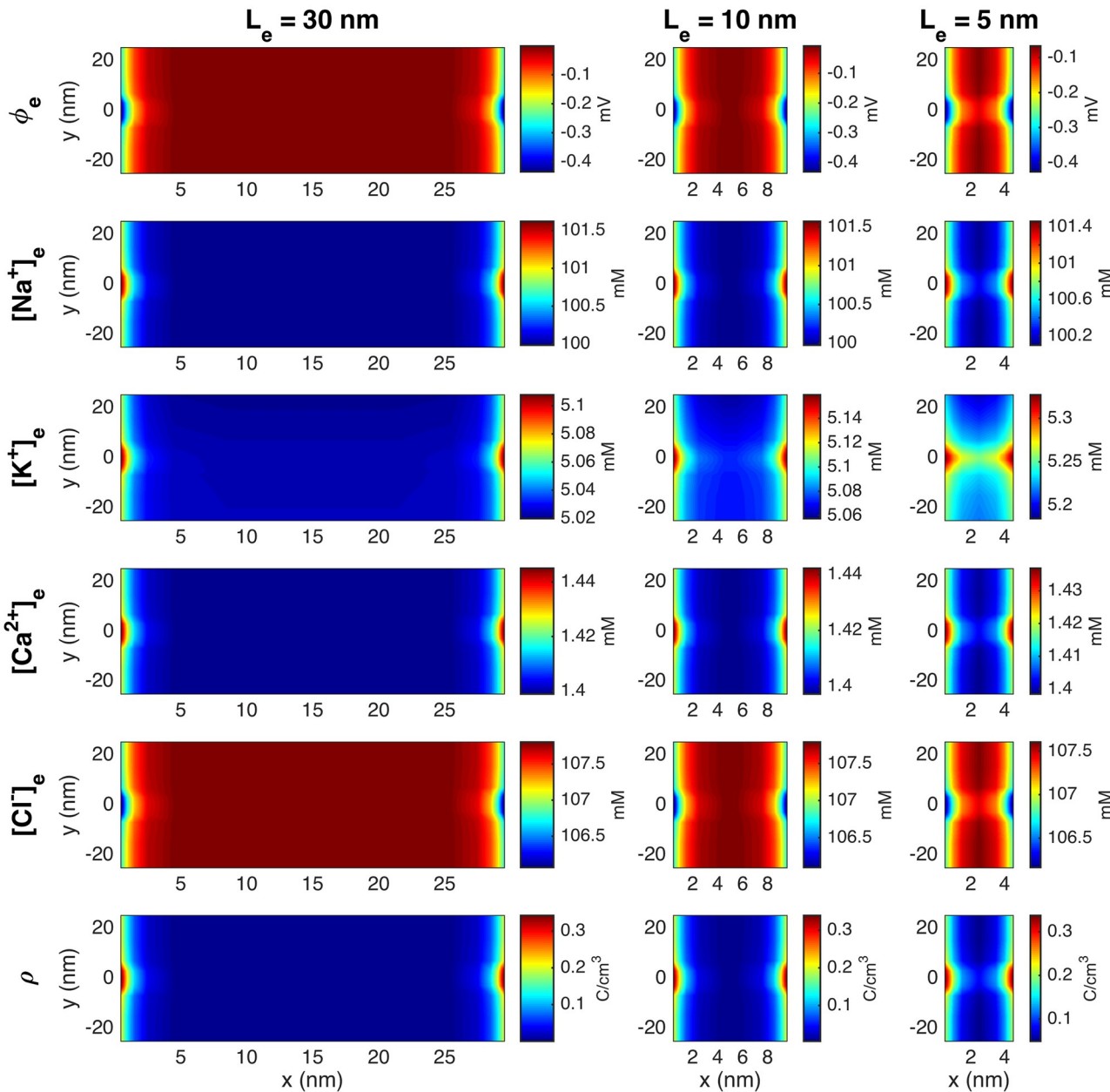

**Fig 6. Stationary solution of the potential, $\phi$, the concentration of Na$^+$, K$^+$, Ca$^{2+}$, and Cl$^-$ ions, and the charge density, $\rho$ in the extracellular space between the two cells in simulations with open K$^+$ channels, but closed Na$^+$ channels.** The width of the extracellular space, $L_e$, is varied in the columns. The plots show the solution in the $(x, y)$-plane for the center of the domain in the $z$-direction at a point in time when steady state is reached. In the $y$-direction, we focus on the 50 nm closest to the K$^+$ channels. The coordinates on the axes are shifted so that $x = 0$ marks the end of the membrane of the pre-junctional cell and $y = 0$ marks the center of the K$^+$ channels. Note that to improve the visibility of the boundary layer, the scaling of the colorbar is different for the different cases.

**3.2.1 A strong and rapid transient emanates from the open Na$^+$ channels.** In Fig 8, we show the solution of a simulation with an extracellular space width of $L_e = 10$ nm and a Na$^+$ channel cluster consisting of 196 channels. We show the solutions in a plane in the $x$- and $y$-directions in the extracellular space close to the Na$^+$ channel cluster at five different points in time. The first column of Fig 8 shows the solution 1 ns after the Na$^+$ channel cluster of the pre-

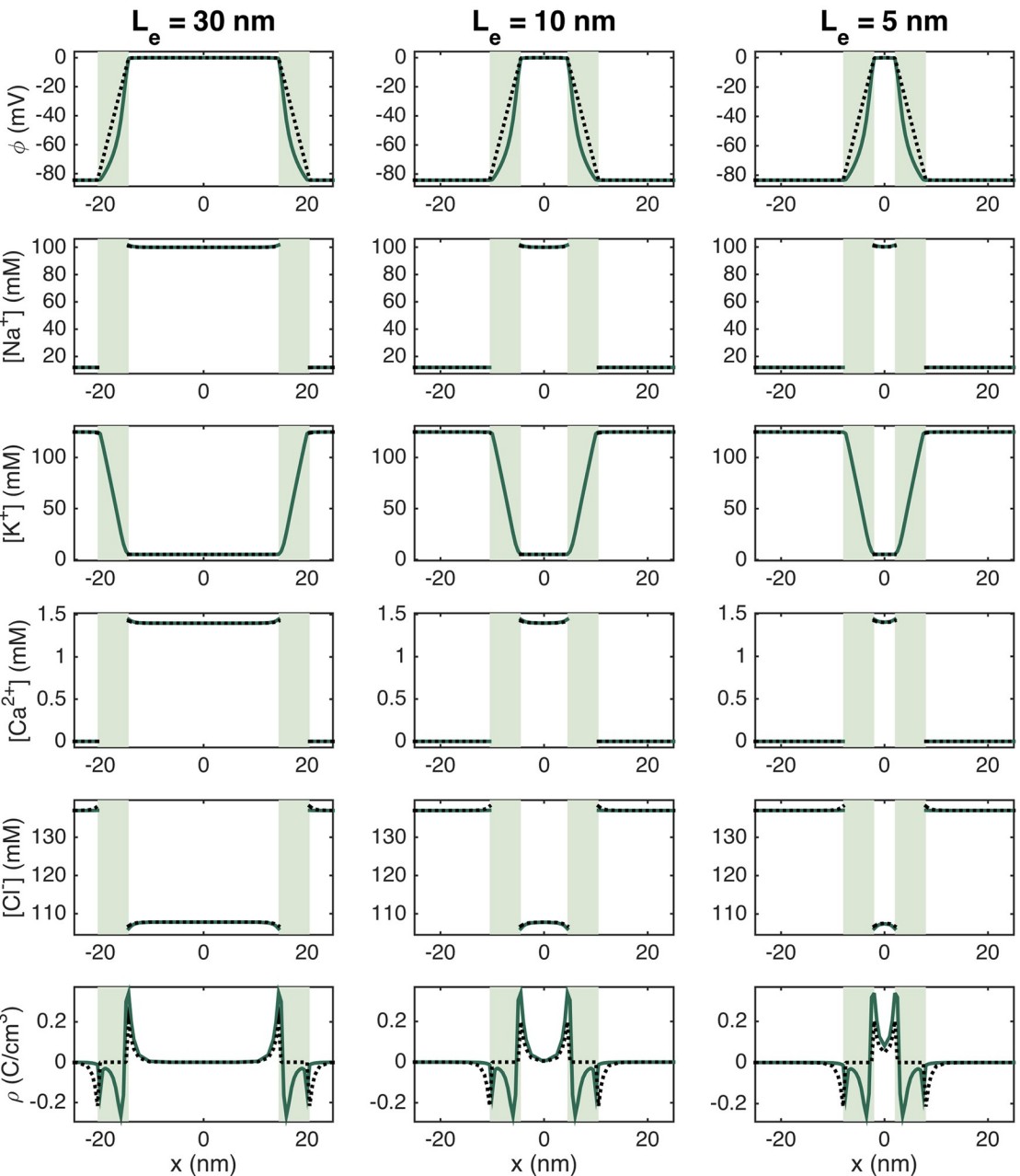

**Fig 7. Stationary solution of the potential, $\phi$, the concentration of Na$^+$, K$^+$, Ca$^{2+}$, and Cl$^-$ ions, and the charge density, $\rho$, along lines in the $x$-direction for open K$^+$ channels and closed Na$^+$ channels.** The full green line represents the solution along a line crossing through the K$^+$ channels and the dotted black line represents the solution along a line about 100 nm below the K$^+$ channel cluster. The light green areas mark the membrane. Note that all ion concentrations are zero in the membrane, except for K$^+$ ions in the K$^+$ channel. The coordinates on the axes are shifted so that $x = 0$ marks the center of the extracellular cleft, and to improve the visibility, the plots only focus on a small part of the intracellular space, closest to the membrane.

junctional cell is opened. We observe that the Na$^+$ concentration is somewhat decreased outside of the channel cluster, as Na$^+$ ions have started to move into the cell through the open channel cluster. The charge density, $\rho$, is also negative outside of the Na$^+$ channel cluster, and the extracellular potential is more negative outside of the channel cluster than at rest.

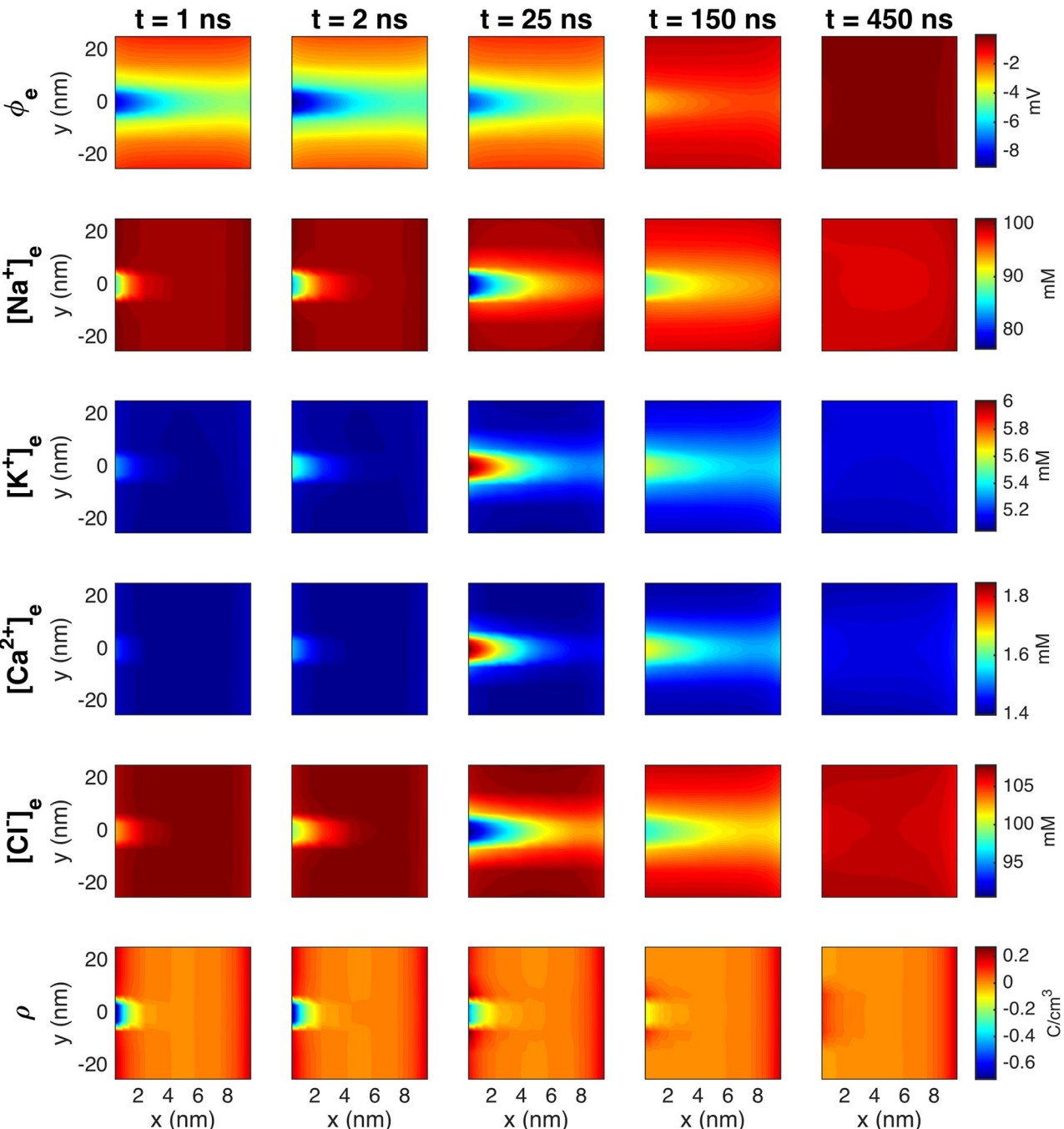

**Fig 8. The PNP model solution in the extracellular space between two cells in a simulation with an Na⁺ channel cluster of 196 channels on the membrane of the pre-junctional cell (the channels are opened at $t = 0$).** The width of the extracellular space, $L_e$, is 10 nm. The plots show the solution in the $(x, y)$-plane for the center of the domain in the $z$-direction at five different points in time (specified in the column titles). In the $y$-direction, we focus on the 50 nm closest to the Na⁺ channel clusters. The coordinates on the axes are shifted so that $x = 0$ marks the end of the membrane of the pre-junctional cell and $y = 0$ marks the center of the Na⁺ channel cluster. Note that the short duration of the dynamics is an artefact caused by the small membrane area associated with the Na⁺ channel cluster in the simulation (see Sections 3.2.3–3.2.5 for more details and an estimation of a more realistic duration).

Furthermore, the $K^+$ and $Ca^{2+}$ concentrations are slightly increased and the $Cl^-$ concentration is slightly decreased in the vicinity of the open $Na^+$ channel cluster.

In the next column, at $t$ = 2 ns, the changes to the ion concentrations are more pronounced and extend further into the extracellular space between the cells. In addition, the extracellular potential is at its most negative, with a value of about −9 mV in an area outside of the open $Na^+$ channel cluster of the pre-junctional cell. In the extracellular space outside of the membrane of the post-junctional cell, the potential is about −5 mV. The charge density outside of the channel cluster appears to be similar to that observed at 1 ns. The next column shows the solutions at the time when the changes to the ion concentrations peak, at $t$ = 25 ns. Now, the ion concentration changes extend even further into the extracellular space. The changes in extracellular potential and the charge density, on the other hand, are decreasing. In the final two columns, the changes in both the extracellular potential, $\rho$ and the ion concentrations decreases. Note that the duration of these dynamics in this simulation might not be physiologically realistic because of the small size of the simulated domain and the small size of the simulated membrane. This is investigated in Sections 3.2.3–3.2.5. In Section 3.2.5 a more realistic duration is estimated.

**3.2.2 The magnitude of the electrochemical wave depends strongly on the number of open sodium channels and the size of the extracellular space.**   Fig 9 shows the extracellular potential, $\phi$, between the two cells at the point in time when the largest deviation from rest occurs after the $Na^+$ channel cluster of the pre-junctional cell has been opened for three different lengths of the extracellular space ($L_e$ = 5 nm, $L_e$ = 10 nm and $L_e$ = 30 nm) and four different sizes of the $Na^+$ channel clusters (see Fig 2). We observe that for all the different $Na^+$ channel cluster sizes, the magnitude (absolute value) of the most negative extracellular potential increases when the width of the extracellular space in decreased. Furthermore, we observe that for $L_e$ = 30 nm or $L_e$ = 10 nm, the magnitude of the extracellular potential is considerably lower outside of the post-junctional cell than outside of the pre-junctional cell (with open $Na^+$ channels), while for $L_e$ = 5 nm, the magnitude of the extracellular potential appears to be quite similar across the extracellular space between the cells. This indicates that the strong negative extracellular potential generated when the $Na^+$ channels open will reach the neighboring cell when the cells are sufficiently close.

We also observe that the magnitude of the extracellular potential increases when the size of the $Na^+$ channel cluster is increased. For $L_e$ = 30 nm, the most negative extracellular potential goes from about −0.5 mV for a cluster of 4 $Na^+$ channels to about −15 mV for 900 channels, and for $L_e$ = 5 nm, the most negative potential goes from about −0.7 mV for 4 $Na^+$ channels to about −30 mV for 900 channels.

Figs 10 and 11 show similar plots of the extracellular $Na^+$ and $K^+$ concentrations. In Fig 10, we observe that the extracellular $Na^+$ concentration is locally decreased outside of the $Na^+$ channel cluster. For $L_e$ = 30 nm, this does not seem to affect the $Na^+$ concentration outside of the post-junctional cell, but for $L_e$ = 5 nm, there seems to be some effect outside of the post-junctional cell in the cases of large $Na^+$ channel clusters. In Fig 11, we observe that the $K^+$ concentration is locally increased outside of the $Na^+$ channel cluster. In addition, we observe a general increase in extracellular $K^+$ concentration as $L_e$ is decreased. Similar plots for the extracellular $Ca^{2+}$ and $Cl^-$ concentrations are found in S1 Appendix.

Fig 12 summarizes the results of adjusting the extracellular width and the number of $Na^+$ channels on the magnitude of changes in the potential and ion concentrations in the extracellular space between the cells. Again, we observe that the magnitude of the negative extracellular potential increases as the distance between the cells ($L_e$) is decreased. The extracellular potential also increases when the number of $Na^+$ channels in the channel clusters increases, and for $L_e$ = 5 nm, the effect is almost identical outside of the post-junctional cell as outside of the pre-

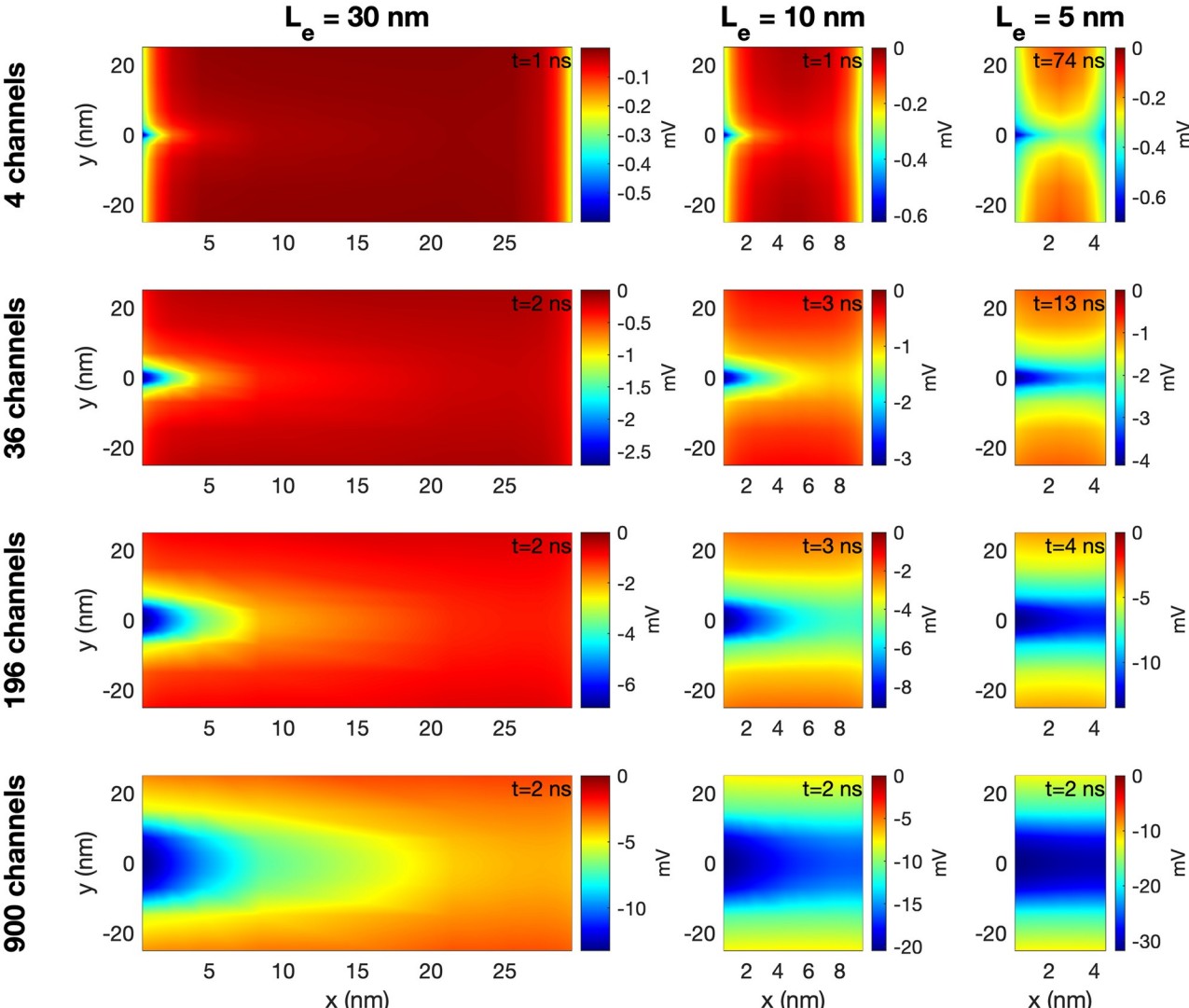

**Fig 9. The potential, $\phi$, in the extracellular space between the two cells in simulations with open Na$^+$ channel clusters on the membrane of the pre-junctional cell.** The width of the extracellular space, $L_e$, and the size of the Na$^+$ channel cluster is varied in the columns and rows of the figure, respectively. The plots show the solution in the $(x, y)$-plane for the center of the domain in the $z$-direction at the point in time when the deviation from rest is largest. This time point (defined as the time after the Na$^+$ channel is opened) is specified in the upper right corner of each plot. In the $y$-direction, we focus on the 50 nm closest to the Na$^+$ channel clusters. The coordinates on the axes are shifted so that $x = 0$ marks the end of the membrane of the pre-junctional cell and $y = 0$ marks the center of the Na$^+$ channel cluster. Note that the scaling of the colorbar is different for the different cases.

junctional cell. For the extracellular widths of $L_e$ = 30 nm and $L_e$ = 10 nm, the extracellular potential is not as negative outside of the post-junctional cell as outside of the pre-junctional cell. Similarly, the ion concentration changes outside of the open Na$^+$ channel cluster of the pre-junctional cell generally seem to increase when the cleft width is decreased or the number of Na$^+$ channels is increased. However, there are some deviations from this general tendency. In particular, the ion concentration changes seem to be similar or even smaller when the number of channels is increased from 196 to 900. In addition, the ion concentration changes outside of the pre-junctional cell seem to be very similar for $L_e$ = 10 nm and $L_e$ = 30 nm. Outside of the post-junctional cell, the ion concentration changes are smaller than outside of the pre-

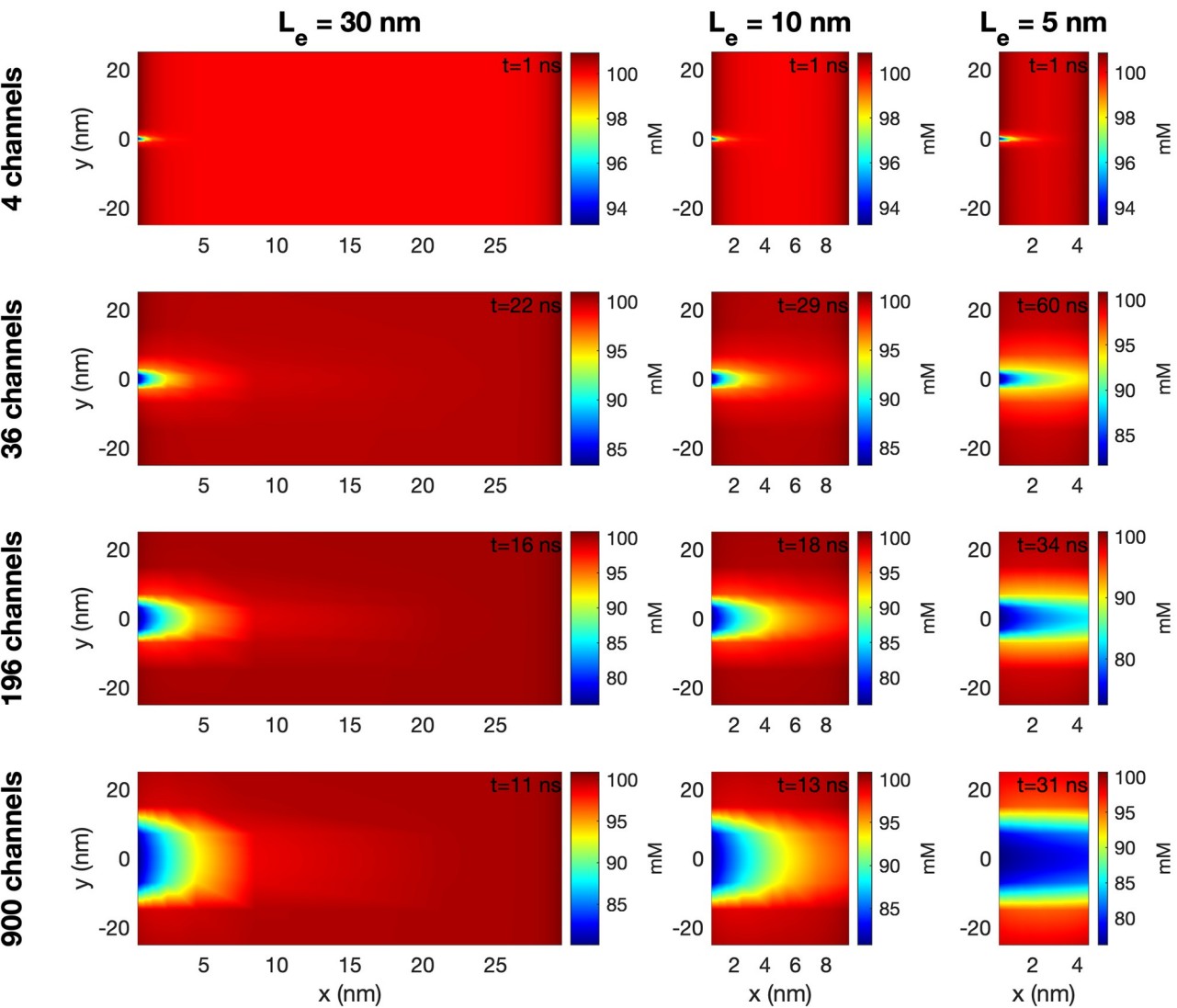

**Fig 10. The Na⁺ concentration in the extracellular space between the two cells in simulations with open Na⁺ channel clusters on the membrane of the pre-junctional cell.** The figure setup is the same as for Fig 9.

junctional cell, and, as expected, the difference between the deviations in ion concentrations outside of the two cells seem to become larger as the cleft width is increased.

**3.2.3 The duration of the electrochemical wave depends on the membrane area included in the simulation.** In the simulations reported above, we observe a strong electrochemical wave emanating from the pre-junctional cell when the Na⁺ channels open. The wave reaches towards the cell on the right-hand side and the strength depends on the number of ion channels available and the width of the extracellular space between the cells (see Figs 9–12). Also, we have seen that the wave is extremely brief. The changes in the potential and the ion concentrations in the simulation reported in Fig 8 last for only about 500 ns. The duration of this wave is critical for understanding ephaptic coupling. If the wave is extremely brief, it is not clear that it will be able to open the Na⁺ channels in the next cell, but if the wave lasts for a sufficiently long time, it may be strong enough to excite the next cell. We therefore need to obtain a firm understanding of the duration of the wave.

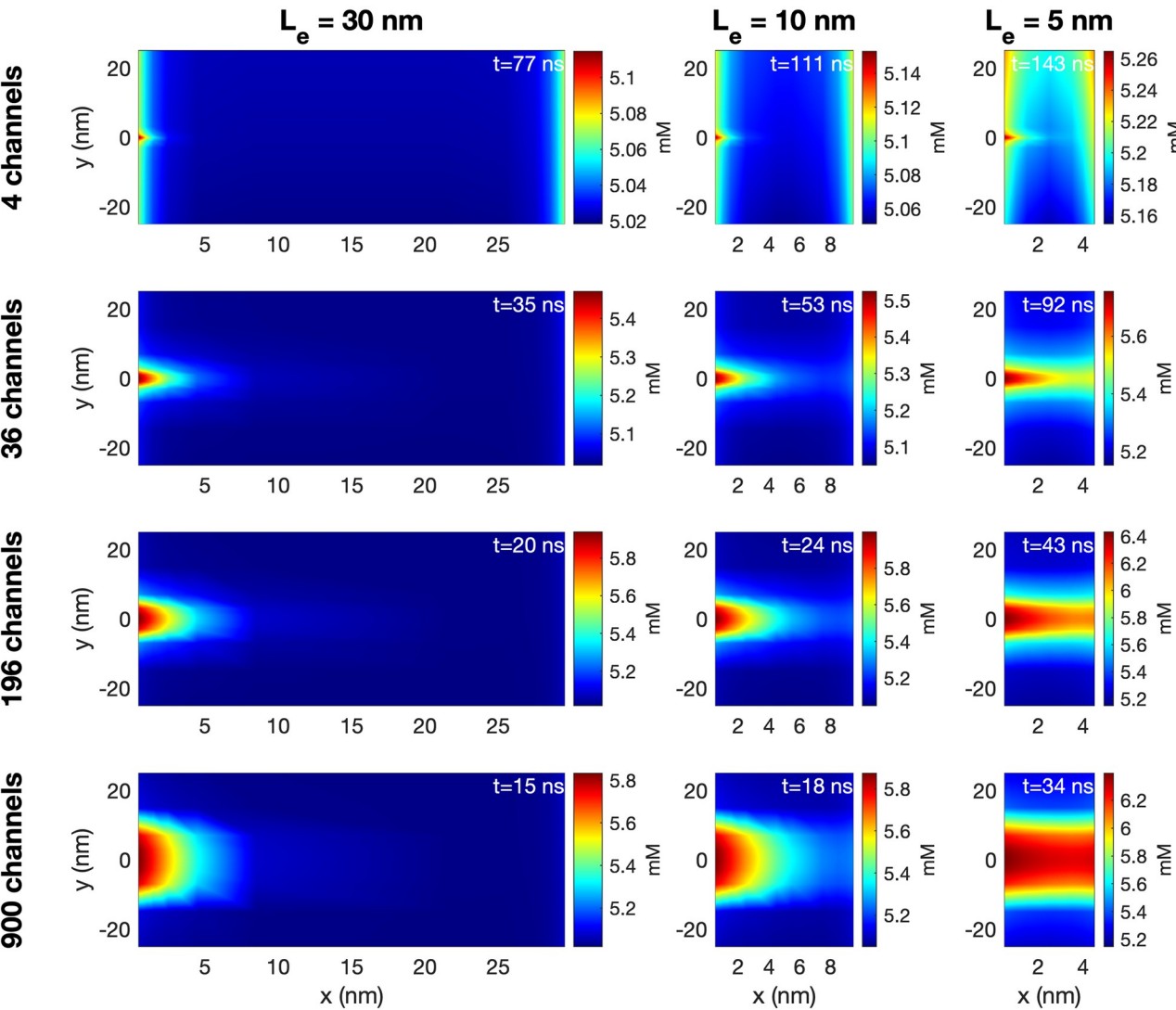

**Fig 11. The K$^+$ concentration in the extracellular space between the two cells in simulations with open Na$^+$ channel clusters on the membrane of the pre-junctional cell.** The figure setup is the same as for Fig 9.

Fig 13 shows the membrane potential, the extracellular potential, the extracellular Na$^+$ and K$^+$ concentrations, the charge density and the total Na$^+$ current in simulations with four different values of $L_y$ and $L_z$. The remaining parameter values are the same in the different simulations. Mimicking inactivation of the sodium channels, the $I_{Na}$ channels are forced to close when the membrane potential reaches 30 mV (resulting in the dents in the solutions observed towards the end of the simulations). The scaling of the time axis is different in the different columns of the figure, and we observe that as the membrane area is increased, the duration of the dynamics is increased.

The results of the simulations are summarized in Fig 14. In the left panel, we observe that the upstroke duration, i.e., the time for the membrane potential to increase from the resting potential to 30 mV appears to be proportional to the membrane area included in the simulation. From Fig 13, we also see that this duration seems to correspond to the duration of the change in extracellular potential and concentrations. The magnitude of the $I_{Na}$ peak, on the

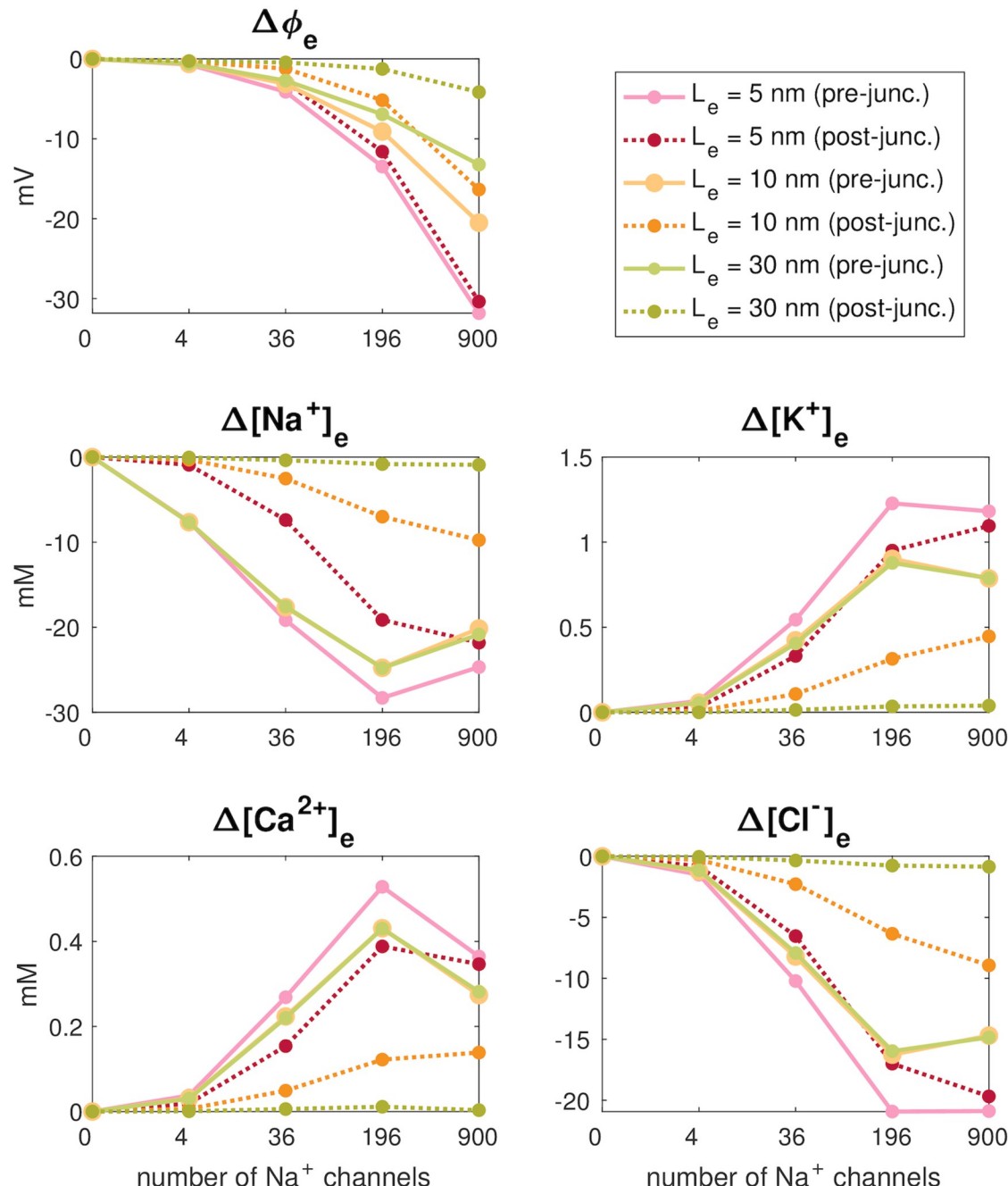

**Fig 12. Largest deviations from rest in the extracellular potential and ion concentrations outside of the Na⁺ channel clusters in simulations with different widths of the extracellular space between the cells, $L_e$, and different sizes of the Na⁺ channel clusters.** The full lines show the solution outside of the Na⁺ channel cluster of the pre-junctional cell, which has open Na⁺ channels. The dotted lines show the solution on the other side of the extracellular space, outside of the post-junctional cell, which has closed Na⁺ channels.

other hand, appears to be very similar for the different membrane areas. In addition, as the membrane area increases, the extracellular potential seems to become a bit more negative, and the minimum value of the Na⁺ concentration decreases and the maximum value of the K⁺ concentration increases.

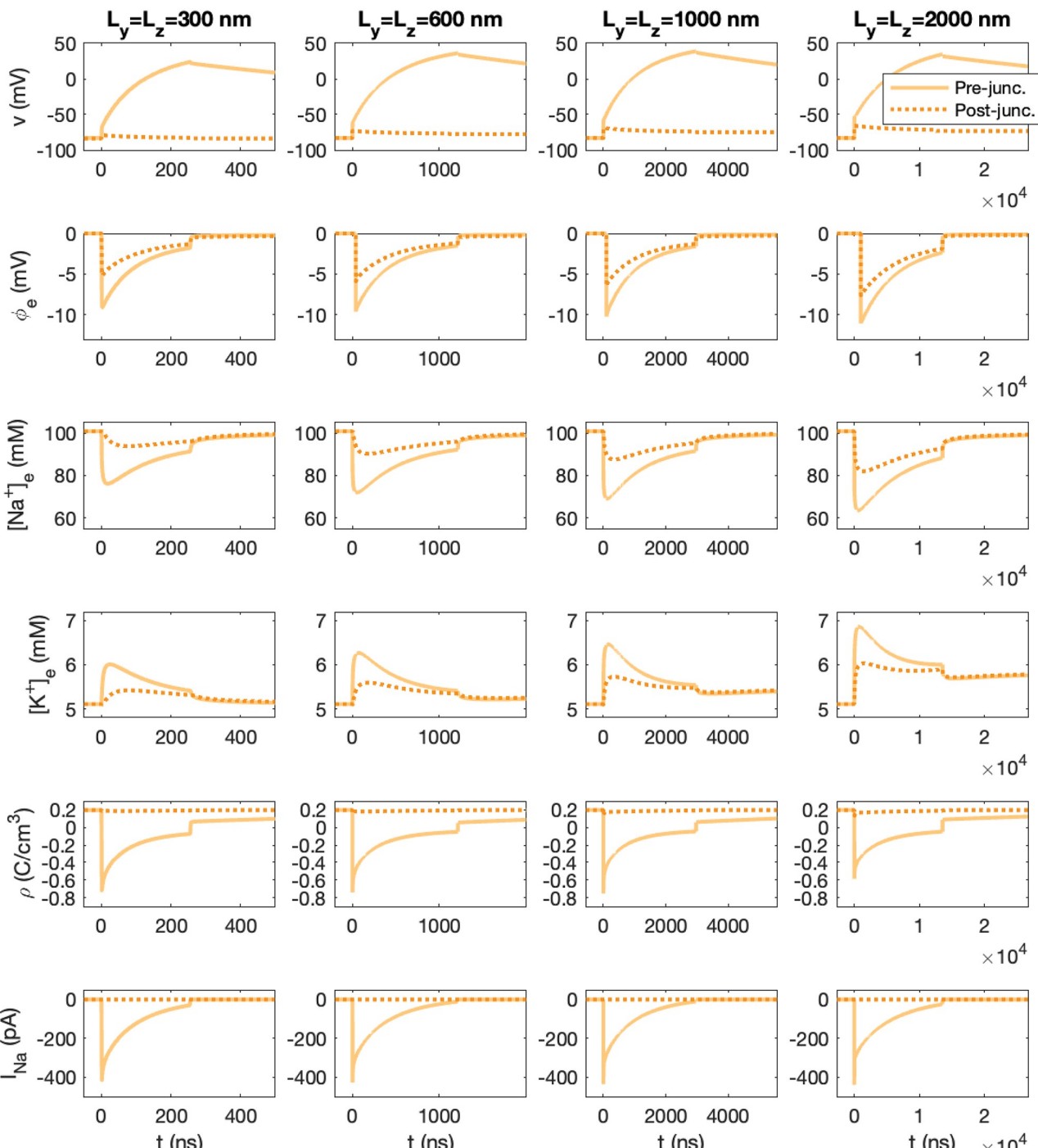

**Fig 13. The membrane potential, $v$, the extracellular potential, $\phi$, the extracellular Na$^+$ and K$^+$ concentrations, the charge density, $\rho$, and the total $I_{\text{Na}}$ current of the pre-and post-junctional cells as functions of time.** The membrane potential and $I_{\text{Na}}$ are measured as described in Fig 5, and $\phi$ and the ionic concentrations are recorded in the first grid point (in the $x$-direction) outside of the center (in the $y$- and $z$-directions) of the Na$^+$ channel clusters. In the first column, the total membrane area included for each cell is 300 nm × 300 nm, and in the next columns, the membrane area is 600 nm × 600 nm, 1000 nm × 1000 nm, and 2000 nm × 2000 nm. In the simulations, the cell distance is $L_e = 10$ nm and the Na$^+$ channel cluster consists of 196 Na$^+$ channels (see Fig 2) and the remaining parameter values are as specified in Tables 1 and 3. The Na$^+$ channels of the pre-junctional cell are opened at $t = 0$, and the Na$^+$ channels of the post-junctional cell are closed in the entire simulations. Note that the scaling of the time axis ($x$-axis) is different in the different columns.

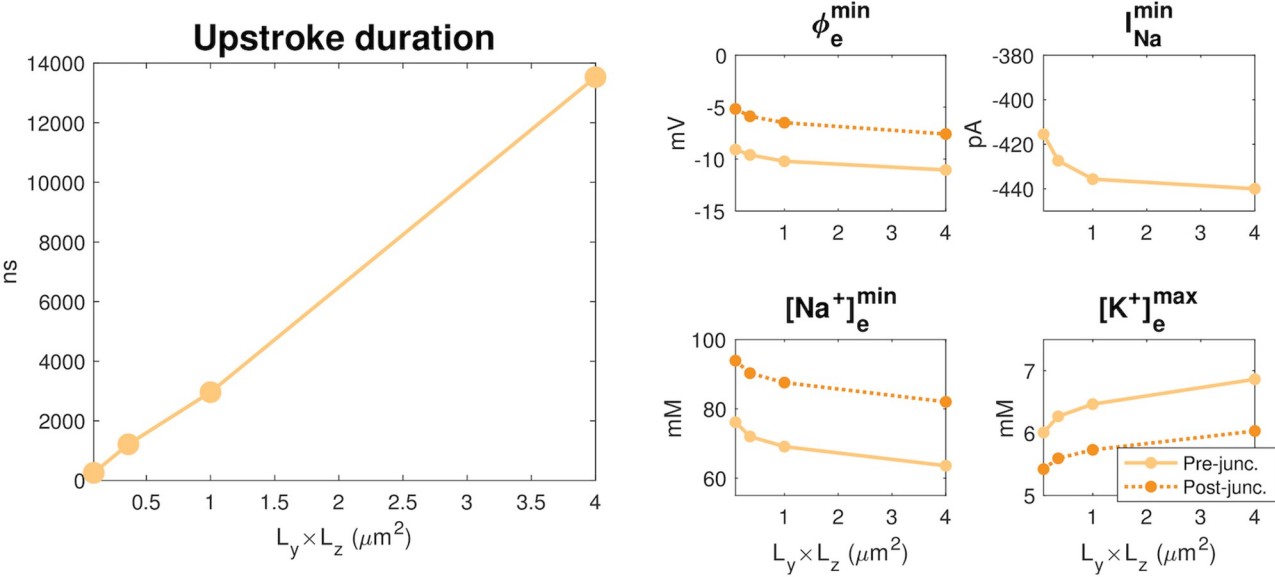

**Fig 14. Summary of the results in Fig 13.** Left panel: The upstroke duration for the membrane potential of the pre-junctional cell as a function of the membrane area $L_y \times L_z$. This upstroke duration corresponds to the time for the membrane potential of the pre-junctional cell to increase from the resting potential to $v = 30$ mV after the $I_{Na}$ channel cluster has been opened. Right panels: The minimum value of $\phi$ and $[Na^+]$ and the maximum value of $[K^+]$ outside of the $Na^+$ channel cluster of the pre-junctional cell (solid lines) and at the level of the post-junctional membrane facing the cluster (dashed lines), as well as the minimum value of $I_{Na}$ (the $I_{Na}$ peak) of the pre-junctional cell as functions of the membrane area.

**3.2.4 Upstroke duration in a simplified model.** To further investigate the relationship between the membrane area included in the simulation and the duration of the upstroke of the membrane potential and the duration of the $I_{Na}$ current, we consider the following classical simplified model for how the membrane potential of excitable cells changes with time as a result of the ionic currents through the ion channels of the membrane:

$$C_m \frac{dv}{dt} = -\sum_i I_i. \tag{21}$$

Here, $C_m$ is the capacitance of the membrane (in pF), $v$ is the membrane potential (in mV), and $I_i$ are currents (in pA) through different types of membrane ion channels. The capacitance, $C_m$, can be expressed as

$$C_m = c_m A_m = \frac{\varepsilon_0 \varepsilon_m}{L_m} A_m, \tag{22}$$

where

$$c_m = \frac{\varepsilon_0 \varepsilon_m}{L_m} \tag{23}$$

is the specific membrane capacitance (in pF/nm²), $\varepsilon_0$ is the vacuum permittivity (in pF/nm), $\varepsilon_m$ is the (unitless) relative permittivity of the medium, $L_m$ is the width of the membrane, and $A_m$ is the area of the membrane (in nm²) [45]. Furthermore, the current through a type of ion channel can be expressed by the simplified model

$$I_i = N_i g_i^0 o_i (v - E_i) = g_i (v - E_i), \tag{24}$$

where $N_i$ is the number of ion channels of type $i$, $g_i^0$ is the conductance through a single open

**Table 4. Parameter values for simplified versions of the $I_{Na}$ and $I_K$ currents of the form (24) fitted to the currents in the PNP simulations in Fig 13.** In these simulations, $N_{Na}$ is 196, $N_K$ is 36 and $o_{Na} = o_K = 1$.

| Parameter | Value |
|---|---|
| $g_{Na}^0$ | $20 \cdot 10^{-3}$ nS |
| $g_K^0$ | $4.0 \cdot 10^{-3}$ nS |
| $E_{NA}$ | 36 mV |
| $E_K$ | −80 mV |

channel of type $i$ (in nS), $o_i$ is the open probability of the channels of type $i$, $E_i$ is the Nernst equilibrium potential of the ion type $i$, and

$$g_i = N_i g_i^0 o_i \tag{25}$$

(in nS) is the total conductance of channels of type $i$ [46]. In our case, we consider two types of ion currents ($I_{Na}$ and $I_K$), and assume that both types of channels are open (i.e., $o_i = 1$). Table 4 reports parameterizations of simplified formulations of these currents of the form (24), fitted to the currents observed in the PNP simulations in Fig 13. Note that since we assume that the open probability of the Na$^+$ channels is 1 ($o_{Na} = 1$) in the simplified model, whereas the Na$^+$ channels close when $v$ reaches a value of 30 mV in the PNP simulations, we only use the simplified model to approximate the PNP model during the upstroke of the action potential, before the Na$^+$ channels close.

In the case of the two currents $I_{Na}$ and $I_K$ of the form (24) with $o_K = o_{Na} = 1$, the simplified model (21) has the analytical solution

$$v(t) = \frac{g_K E_K + g_{Na} E_{Na} + g_{Na}(E_K - E_{Na})e^{-\frac{(g_K + g_{Na})t}{A_m c_m}}}{g_K + g_{Na}}, \tag{26}$$

assuming that $v(0) = E_K$. In Fig 15, we compare the analytical solution of the simplified model to the solution of the PNP model for the simulations considered in Fig 13. We observe that the simplified model seems to be a relatively good approximation for the upstroke of the PNP solution for different sizes of the membrane area.

From the formula (26), the duration from $t = 0$ (i.e., when the Na$^+$ channels open) to when the membrane potential has reached the value 30 mV is given by

$$t = A_m c_m \frac{\ln\left(\frac{g_{Na}(E_K - E_{Na})}{(g_K + g_{Na})\cdot(30 \text{ mV}) - g_K E_K - g_{Na} E_{Na}}\right)}{g_K + g_{Na}}. \tag{27}$$

In this formula, it is clear that the time for the membrane potential to reach 30 mV depends linearly of the membrane area included, $A_m$. In the rightmost panel of Fig 15, we observe that this approximation of the duration of the upstroke is in very good agreement with the durations observed in the PNP simulations.

**3.2.5 The extracellular wave will last for about 0.4 ms.** In the simulations reported in Fig 13, we consider 196 open Na$^+$ channels, and obtain a peak total $I_{Na}$ current of about −400 pA. For a membrane area of 300 nm × 300 nm (like in Fig 8), the peak current density for the $I_{Na}$ current is $\frac{-400\text{pA}}{(300\text{nm})^2} \approx -430,000 \ \mu A/cm^2$. For comparison, in typical models of the cardiac action potential, the peak current density is about −400 $\mu A/cm^2$ (see, e.g., [46, 47]). This indicates that the Na$^+$ channel cluster of 196 channels should have been associated with a membrane area that is about 1000 times larger than the 300 nm × 300 nm area used here in

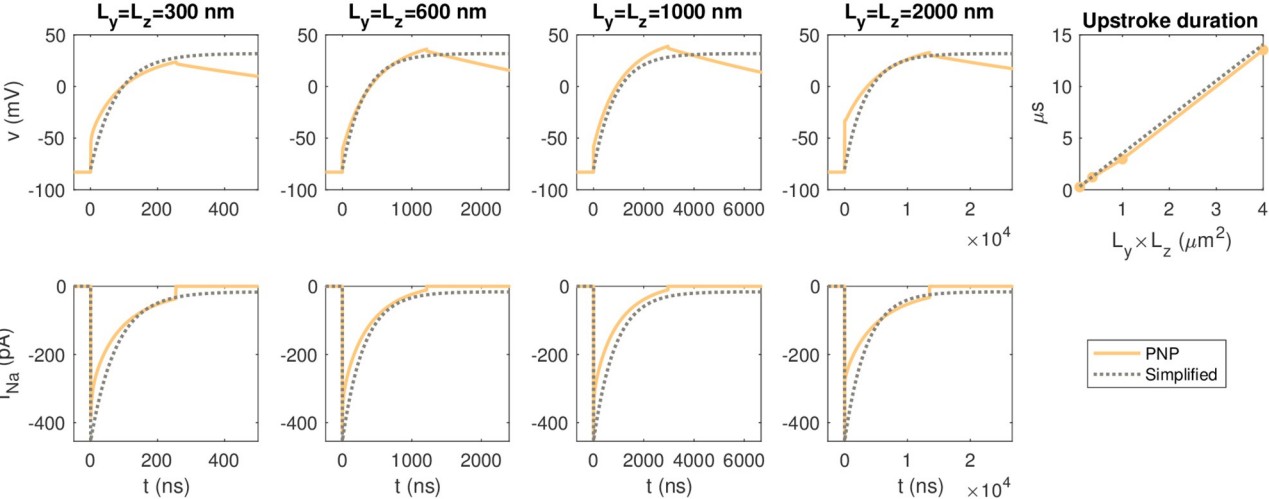

**Fig 15. The membrane potential, $v$, and the total $I_{Na}$ current of the pre-junctional cell as functions of time in the PNP simulations reported in Fig 13 and in the solution of the simplified model (21)–(26).** The $Na^+$ channel cluster consists of 196 open $Na^+$ channels. In the PNP simulations, the membrane potential and $I_{Na}$ are measured as described in Fig 5, the cell distance is $L_e = 10$ nm, and the remaining parameter values are as specified in Tables 1 and 3. In the simplified model, the parameter values are as specified in Tables 1, 3 and 4. Note that the scaling of the time axis ($x$-axis) is different in the different columns. The rightmost column reports the time for the membrane potential to increase from the resting potential to $v = 30$ mV as a function of the membrane area in the PNP simulations and as computed by the analytical formula (27). Note that since we assume that the open probability of the $Na^+$ channels is 1 ($o_{Na} = 1$) in the simplified model, whereas the $Na^+$ channels close when $v$ reaches a value of 30 mV in the PNP simulations, we only use the simplified model to approximate the PNP model during the upstroke of the action potential, before the $Na^+$ channels close.

order to represent a realistic duration of the dynamics. For example, an area of 10000 nm × 10000 nm could be suitable. We are not, however, currently able to perform such large simulations due to the computational load of running long simulations.

On the other hand, since the simplified model (22) provided results that almost coincide with the PNP results, and the linear relation between the area of the membrane and the duration of the wave is approximated very well, we can use the simplified model to estimate the duration of the wave for any membrane area. By using (27) and the parameters given in Table 4, we find that the wave will last for about 0.35 ms for the 10000 nm × 10000 nm membrane area. This is most likely sufficient to initiate a depolarization of the next cell when the wave reaches across the extracellular space if the magnitude of the wave is sufficiently large. Adjusting the membrane area similarly to obtain a current density of about $-400 \mu A/cm^2$ for the other $Na^+$ channel cluster sizes, this duration will be the same for the different cluster sizes.

We also note that the results in Figs 13 and 14 indicate that the magnitude of the change in the extracellular potential and the ion concentrations increases as the membrane area is increased. From these observations, we expect that the changes in potential and concentrations in a simulation with a more realistic membrane area would be at least as large as those observed in our small simulations with a membrane area of 300 nm × 300 nm (see Figs 9–12), and possibly even larger. Nevertheless, this estimate is associated with a level of uncertainty, since the membrane areas currently investigated are considerably smaller than the membrane area needed for realistic durations of the dynamics and because the simplified model is not suitable for estimating changes in the extracellular potential or ion concentrations following from an adjusted membrane area.

**Table 5. The maximum of the absolute value of the derivative of the potential and the concentrations with respect to time and with respect to $x$ in the PNP simulation reported in Fig 8.** In this simulation, the K$^+$ channels are open for both cells, and a Na$^+$ channel cluster consisting of 196 channels are opened for the pre-junctional cell. The extracellular space width is $L_e$ = 10 nm. As a comparison, the transmembrane electric field is in the range of 15 mV/nm for a membrane potential of $-90$ mV.

| | $\max\left|\frac{\partial}{\partial t}\right|$ | $\max\left|\frac{\partial}{\partial x}\right|$ |
|---|---|---|
| $\phi$ | 26 mV/ns | 51 mV/nm |
| $c_{Na^+}$ | 13 mM/ns | 22 mM/nm |
| $c_{K^+}$ | 0.3 mM/ns | 28 mM/nm |
| $c_{Ca^{2+}}$ | 0.05 mM/ns | 0.06 mM/nm |
| $c_{Cl^-}$ | 5.5 mM/ns | 6 mM/nm |

### 3.3 The gradients of the solutions of the PNP equations

As mentioned above, the gradients involved in the solution of the full PNP model are substantial. The maximum gradients in an example simulation are listed in Table 5. These gradients imply that extremely fine meshes and time steps must be applied and this may be the reason why the model is often simplified by assuming electroneutrality. Numerical instabilities arise if the spatial or temporal resolution are too coarse, see, e.g., [11].

## 4 Discussion

Using the Poisson-Nernst-Planck equations, we developed a computational modelling framework to investigate electric potentials and ion concentration changes in the vicinity of ion channel clusters in two opposing cell membranes, as found, e.g., in cardiac intercalated discs. For the present study, we focused on Na$^+$ and K$^+$ channel clusters that vary in size, corresponding to different numbers of channels in the clusters. To develop the model and solve the mathematical problem, a very fine temporal and spatial discretization was required, leading to a very large computational effort. We limited the computational expenses by reducing the represented membrane areas and modelling only a small part of the intercalated disc membranes. To additionally reduce the number of nodes, an adaptive mesh was used for spatial discretization.

When modelling complex dynamical processes in a small part of two adjacent cells, the choice of appropriate boundary conditions is challenging and of great importance. Therefore, we have tested several alternatives before defining the most suitable boundary conditions for our problem. If only no-flux Neumann boundary conditions are applied on intracellular and extracellular domains, neither current nor ions can enter or exit the modelled region. It is worth noting that Neumann boundary conditions are compatible with the Poisson equation, but corresponds to an unrealistic sealing off of the extracellular intercalated disc domain from the extracellular bulk space. With the aim to simulate the junction between the intercalated disc and bulk space, a Dirichlet condition for ion concentrations and zero potential was therefore applied at the periphery of the extracellular domain. To ensure the formation of the Debye layers close to the cell membranes at the periphery of the simulated extra- and intracellular domains, this Dirichlet condition was applied only in the central part of the extracellular boundary surface (see Fig 3). Given that only a small part of the intercalated disc was modeled, a combined choice of Dirichlet and Neumann boundary conditions is, in our opinion, the most appropriate approach to realistically represent cardiomyocytes joined by an intercalated disc.

### 4.1 Formation of Debye layers

The first main finding in our simulations is the clear and distinct formation of boundary layers, known as Debye layers, for the extracellular potential, the charge density, $\rho$, and the concentrations of $Na^+$, $K^+$, $Ca^{2+}$ and $Cl^-$ when, in a first step, only $K^+$ channels are open and the resting state of the model is reached. These Debye layers arise naturally from Gauss' law of electrostatics [48] and the dielectric property of the membrane. For the small membrane area applied in our simulations, these Debye layers form at the microsecond timescale (see Fig A in S1 Appendix). However, the time scale associated with the formation of the Debye layers likely depends on the membrane area associated with the $K^+$ channels (as observed for the duration of the dynamics following $Na^+$ channel opening in Section 3.2.3; Fig 13.)

### 4.2 Extracellular ionic dynamics

Our second main finding was obtained in the second part of the study, when the $Na^+$ channels in the cluster were opened. The simulations show that near the opening $Na^+$ channel cluster, large concentration changes occur. When $Na^+$ ions enter the cytoplasm through open $Na^+$ channels, the extracellular $Na^+$ concentration at the inlet of the channel cluster decreases. With the large extracellular $Na^+$ depletion resulting from the influx of $Na^+$ into the pre-junctional (left) cell, a strong negative extracellular potential builds up in the extracellular cleft. Concomitantly, in the extracellular domain, $K^+$ ions and $Ca^{2+}$ ions are attracted by the electrochemical gradient towards the $Na^+$ channel cluster, while $Cl^-$ anions are repelled away from it. Indeed, near the extracellular side of the $Na^+$ channel cluster, $[K^+]$ and $[Ca^{2+}]$ transiently rise while $[Cl^-]$ decreases (see Fig 8). Thus, substantial concentration gradients are generated. In the simulation shown in Fig 8, the large ion concentration gradients dissipate quickly and almost completely on a time scale of $\sim$500 ns, however this time scale appears to be proportional to the membrane area associated with the $Na^+$ channel cluster (see Figs 13–15), and a more physiologically realistic duration of the dynamics is estimated to be about 0.4 ms (see Section 3.2.5). More specifically, the changes in the extracellular potential, the charge density and the ionic concentrations in the cleft all seem to last for about as long as the duration of $I_{Na}$ current, which self-limitates as the membrane potential, $v$, approaches the Nernst equilibrium potential for $Na^+$, $E_{Na}$ (see Section 3.2.4).

It should be noted that even though the charge density, $\rho$, is non-zero close to the membrane in our simulations, and in particular close to the open $Na^+$ channel cluster during the upstroke of the membrane potential (see, e.g., Figs 6 and 8), we believe that for larger scale models (e.g., individual cardiomyocytes), electroneutrality is a reasonable and necessary assumption to reduce the large computational effort, as was done in other modeling frameworks examining biological membranes and channels, such as the Kirchhoff-Nernst-Planck formalism [9, 12].

### 4.3 Impact of $Na^+$ channel cluster size and cleft width

Our third main finding is that an increasing number of $Na^+$ channels in a cluster and reduced extracellular cleft width lead to more negative extracellular potentials in the cleft. The number of $Na^+$ channels was modelled by an increased $Na^+$ channel cluster size. The question arises whether the size or the number of $Na^+$ channels leads to the more negative extracellular potential in the cleft. According to simulations with the computational model of the intercalated disc by Hichri et al. [21], when the number of $Na^+$ channels at the intercalated disc membrane remains constant but the cluster size is decreased, the extracellular potential becomes more negative and thus ephaptic effects are accentuated. Therefore, we conclude that the number of $Na^+$ channels rather than the cluster size is the key factor in our simulation results. Moreover,

the reduction of the extracellular cleft width (from 30 nm to 5 nm), leading to more negative extracellular potentials in the cleft is consistent with previously published computational studies at the cellular level [3, 9, 21]. With decreasing cleft width, negative extracellular potentials in the local neighborhood of the $Na^+$ channel cluster of the left cell are transmitted to the other side of the cleft (close to the post-junctional cell), where it is likely to affect $Na^+$ channel gating, thus forming the basis for ephaptic coupling.

## 4.4 Limitations and perspectives

The strength of our modelling framework is the full incorporation of the Poisson-Nernst-Planck equations at the level of ion channel clusters. However, during the development of the model, some highly simplifying assumptions were necessary.

A first simplifying assumption was taken at the level of the ion channel pores, in terms of the free energy profile that a permeating ion is subjected to as it passes through a channel. This energy profile is determined by the distribution of charges within the protein forming the channel, especially in the selectivity filter, which is typically lined up with amino acid residues bearing a charge that is opposite to that of the permeating ion [49]. As the background charge density $\rho_0$ of ion channels (e.g., $Na^+$ and $K^+$ channels) is not precisely known, we assumed for our main results a linear $\rho_0$ profile from the intracellular to the extracellular part of the $Na^+$ and $K^+$ channels. However, as shown in S1 Appendix, different concentration and $\rho_0$ profiles (i.e., an abrupt discontinuity in the center of the channel) affect permeation [49], and thus the simulation results. Moreover, ion channel permeation kinetics are strongly influenced by the dehydration and rehydration of permeating ions inside the channels, which further affect the free energy profile [49]. We did not incorporate these channel features into our model. Another aspect that we did not consider in our framework is the negative charge at the surface of the phospholipid bilayer membrane [49], that may have an additional effect on the simulation results. However, the incorporation of such details into the model was not in the scope of our study. A full model integrating ion channel permeation kinetics would require a mixed framework with molecular dynamics of channel permeation, which would add complexity and computational effort to our already computationally intensive modelling framework.

Because our simulations corresponded to nano- and microsecond time scales, we also assumed in our model that clustered ion channels open at the same time, while in reality they exhibit asynchronous stochastic openings and closings resulting from voltage gating. However, for $Na^+$ channels, this gating occurs at the much longer millisecond scale. In future work, we plan to incorporate the gating of channels, which will allow the examination of action potential propagation via ephaptic coupling. The incorporation of channel gating will provide insight into possible cooperative gating within the channel clusters [50, 51].

Providing direct evidence of ephaptic coupling is difficult because it is technically challenging to measure extracellular potentials in intercalated disc clefts that are only a few tens of nanometers wide. Thus, for the moment, experiments provide only indirect evidence. For example, in [21], the authors examined in voltage clamp experiments how restricting the extracellular space near a cell expressing voltage-gated $Na^+$ channels modifies the resulting $Na^+$ current. It was found that at step potentials just above the excitation threshold, the $Na^+$ current was increased, whereas at step potentials far above threshold, the $Na^+$ current was decreased. Both phenomena are compatible with ephaptic effects and were in line with modeling predictions.

An interesting method to measure extracellular potentials, called *diamond voltage imaging*, has been recently developed, see [52]. This technology allows measuring extracellular potentials in an electrolyte solution at high spatial resolution, and it may thus open perspectives to

provide direct extracellular measurements in narrow extracellular domains. Nanowires that could be inserted into narrow clefts may represent another option.

Chemical or optogenetic reporters of electrical potential or even of ion concentrations that could be directly targeted to the intercalated disc may represent a further interesting approach in the future. However, at the present time, optogenetic reporters exhibit kinetics that are still too slow to permit the detection of rapid potential or ion concentration changes, which, as indicated by our present study, may occur already at the nano- and microsecond scales. Chemical probes are probably more appropriate, but the development of such probes will undeniably by a major challenge. Hence, for the time being, modeling and simulations represent a useful and important approach to gain insight into the behavior of ion channels, electric potentials and ion concentrations in biological nanodomains.

## 5 Conclusion

In conclusion, our new computational model allows a detailed investigation of dynamic ion concentration changes around single $Na^+$ channel and $K^+$ channel clusters in a small fraction of two opposing intercalated disc membranes. The features of the full Poisson-Nernst-Planck modelling framework can be in the future incorporated into larger scale models. Our simulation results highlight the importance of incorporating dynamic ion concentration changes into existing models to obtain a full picture of the interaction between ion channels, ion concentrations, and electric potentials at the spatial and temporal nanoscales. Particularly, this will be crucial to gain further insights into the mechanisms of ephaptic coupling as a complementary mechanism to gap junctional coupling for action potential propagation in the heart. This knowledge may be useful, in the future, to develop more suitable treatments for cardiac arrhythmias.

## Supporting information

**S1 Appendix. Supplementary information.**
(PDF)

**S1 Data. Underlying numerical data for figures.** Excel spreadsheet containing the underlying numerical data of Figs 12 and 14.
(XLSX)

## Author Contributions

**Conceptualization:** Karoline Horgmo Jæger, Ena Ivanovic, Jan P. Kucera, Aslak Tveito.

**Data curation:** Karoline Horgmo Jæger.

**Funding acquisition:** Aslak Tveito.

**Investigation:** Karoline Horgmo Jæger, Ena Ivanovic, Jan P. Kucera, Aslak Tveito.

**Methodology:** Karoline Horgmo Jæger, Aslak Tveito.

**Project administration:** Aslak Tveito.

**Software:** Karoline Horgmo Jæger.

**Visualization:** Karoline Horgmo Jæger.

**Writing – original draft:** Karoline Horgmo Jæger, Ena Ivanovic, Jan P. Kucera, Aslak Tveito.

**Writing – review & editing:** Karoline Horgmo Jæger, Ena Ivanovic, Jan P. Kucera, Aslak Tveito.

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
