## [Decision Letter · Decision Letter 0]

23 Nov 2022

Dear Dr. Jæger,

Thank you very much for submitting your manuscript "Nano-scale solution of the Poisson-Nernst-Planck (PNP) equations in a fraction of two neighboring cells reveals the magnitude of intercellular electrochemical waves" for consideration at PLOS Computational Biology. As with all papers reviewed by the journal, your manuscript was reviewed by members of the editorial board and by several independent reviewers. The reviewers appreciated the attention to an important topic. Based on the reviews, we are likely to accept this manuscript for publication, providing that you modify the manuscript according to the review recommendations.

In addition to incorporating changes based on the reviewers' comments, there was an editorial concern. This paper makes important advances from a numerical methods perspective, but very little attention is made to how these predictions may be validated experimentally or even used to guide future experiments. The discussion should address this as appropriate.

Sincerely,

Jeffrey J. Saucerman

Academic Editor

PLOS Computational Biology

Daniel Beard

Section Editor

PLOS Computational Biology

Reviewer's Responses to Questions

**Comments to the Authors:**

Reviewer #1: In this study by Jaeger and colleagues, the authors present a thorough analysis of the Poisson-Nernst-Planck (PNP) equations to model electrochemical dynamics in a domain representing the narrow separation between two adjacent cardiomyocytes. These dynamics have important implications for the feasibility and role of so-called ephaptic coupling, a mechanism for cell-cell electrical transmission governed by extracellular potential hyperpolarization, which in turn can drive activation of sodium channels in the post-junctional cell.

This investigation required the development of highly complicated and technically challenging numerical methods and simulation approaches. Overall I find the manuscript to be well-written and the study rigorously performed. I have a few comments for the authors, which are mostly minor and mostly serve to improve the presentation of the results.

1. First, in Figure 14, can you provide the corresponding values in the right panel plots for the right cell? In particular phi_e_min would be relevant to understand how hyperpolarized the extracellular potential is adjacent to the Na+ channels on the right cell membrane.

This detail in general is relevant to my largest question. Is the hyperpolarization in the extracellular space adjacent to the right cell sufficient to activate Na+ channels on the right cell membrane in physiological conditions? The trend that phi_e_min decreases (becomes more negative) as the membrane area increase suggests that the hyperpolarization would be larger in a realistic area (as for the upstroke duration), but the trend is not as clearly linear as the upstroke duration. Can the simplified model be applied to estimate the phi_min_e for a larger area? If not, can the authors comment on this more broadly - even if it is speculation - as to their expectations?

2. Regarding the description of the charge density profile near the K+ channel in Figure 7 (bottom row), even after reading the description in the Supplement, I was a bit confused, and I think the description could be a bit more clear.

The authors state that the complex charge density profile arises due to small deviations from the linear profile used as [K+] in the initial conditions, but would it be more accurate to describe the complex profile as due to the channel [K+]'s deviation from a linear profile at steady-state (not initial conditions)?

As I understand, the assumption of a linear profile in [K+] for initial conditions also results in a linear profile in rho_0. Since the steady-state K+ in the channel deviates from the rho_0 profile, the complex rho profile emerges.

3. Line 151: The authors' state "After the simulation in Part 1 seems to have reached a steady state, ..." please be more specific as to what was considered to determine if the Part 1 system had reached steady state.

4. In Figure 9, please comment and specify the value of the time associated with each snapshot. This would enable commenting on how the number of channels and extracellular space width impact the time that the greatest cleft polarization occurs, which in turn may relate to the efficacy of ephaptic transmission.

5. In Figure 15, top row, was the the voltage-dependence of Na+ channel closure incorporated into the simplified model? I would expect that the voltage traces would return back to EK, when the Na+ channels are closed but instead it reaches a plateau.

Minor: I leave this to the authors' discretion, but the use of 'left' and 'right' cell seems a bit arbitrary, and something such as pre- and post-junctional or 'upstream' or 'downstream' could be more appropriate.

Minor: Provide a definition and numerical value fo beta_Na+ in Eqn. 20.

Minor: Can you produce a summary plot comparable to Figure 12 for the extracellular ion concentrations as well? This concise summary would be helpful.

Minor: Typo on line 481 "maxmimum"

Reviewer #2: In this manuscript, Horgmo Jæger et al. present a computational modelling framework to investigate electric and ionic dynamics in the vicinity of ion channel clusters in two opposing cell membranes, focusing on Na+ and K+ channels. Use of Poisson-Nernst-Planck equations leads to a huge computational load, which authors reduced by clever and well-justified approaches. The study provides great insights to 1) formation of boundary layers, i.e. Debye layers, 2) extracellular or in-cleft ionic dynamics, and 3) impact of Na+ channel cluster and cleft width on the ephaptic coupling. The manuscript is well-structured, and written in very fluent English.

Minor comments and stylistic suggestions:

- line 40: “perturbing the dynamics”  “perturbing its dynamics”

- line 48: “the upstroke takes”  “the upstroke of the AP takes”

- line 50: “to represent that type of areas in”  “to represent that area scale in”

- Table 1: The chosen initial value for intracellular Na+, 12 mM, is rather high. At least to me, it sounds like a value typical for rodent cardiomyocytes, or maybe failing human cardiomyocytes. In healthy human cardiomyocytes, something like 8 mM would be more realistic. This is just a general note. I do realise that the authors do not explicitly state that their modelling effort is in the context of human cardiomyocytes, even though refs 41 and 42 indicate towards that. Furthermore, I would not expect a change from 12 to 8 mM to affect the findings in any relevant way.

- line 193: “The concentrations are in this area ﬁxed at”  “The concentrations in this area are ﬁxed at”

- line 343: “reach the the neighboring”  “reach the neighboring”

- line 482: “extremely ﬁnes meshes”  “extremely ﬁne meshes”

- Discussion: To improve readability, I suggest that the authors consider using subheadings: 4.1 Formation of Debye layers, 4.2) Extracellular ionic dynamics, 4.3 Impact of Na+ channel cluster and cleft width, and 4.4 Limitations and perspectives.

**Have the authors made all data and (if applicable) computational code underlying the findings in their manuscript fully available?**

Reviewer #1: **No: **The authors note that code will be provided at publication.

Reviewer #2: Yes

PLOS authors have the option to publish the peer review history of their article (what does this mean?). If published, this will include your full peer review and any attached files.

Reviewer #1: No

Reviewer #2: **Yes: **Jussi T. Koivumäki

Figure Files:

Data Requirements:

Reproducibility:

References:

---

## [Editor Report · Decision Letter 1]

23 Jan 2023

Dear Dr. Jæger,

We are pleased to inform you that your manuscript 'Nano-scale solution of the Poisson-Nernst-Planck (PNP) equations in a fraction of two neighboring cells reveals the magnitude of intercellular electrochemical waves' has been provisionally accepted for publication in PLOS Computational Biology.

Best regards,

Jeffrey J. Saucerman

Academic Editor

PLOS Computational Biology

Daniel Beard

Section Editor

PLOS Computational Biology

---

## [Editor Report · Acceptance letter]

10 Feb 2023

PCOMPBIOL-D-22-01327R1 

Nano-scale solution of the Poisson-Nernst-Planck (PNP) equations in a fraction of two neighboring cells reveals the magnitude of intercellular electrochemical waves

Dear Dr Jæger,

I am pleased to inform you that your manuscript has been formally accepted for publication in PLOS Computational Biology. Your manuscript is now with our production department and you will be notified of the publication date in due course.

With kind regards,

Zsofia Freund
